# The MiDAC histone deacetylase complex is essential for embryonic development and has a unique multivalent structure

Robert E. Turnbull [1,2,9], Louise Fairall[1,2,9], Almutasem Saleh[1,2,5,9], Emma Kelsall[2,6,9], Kyle L. Morris[3,7], T. J. Ragan [1], Christos G. Savva[1], Aditya Chandru[2,8], Christopher J. Millard [1,2], Olga V. Makarova[2], Corinne J. Smith [3], Alan M. Roseman[4], Andrew M. Fry[2], Shaun M. Cowley [2]✉ & John W. R. Schwabe [1,2]✉

MiDAC is one of seven distinct, large multi-protein complexes that recruit class I histone deacetylases to the genome to regulate gene expression. Despite implications of involvement in cell cycle regulation and in several cancers, surprisingly little is known about the function or structure of MiDAC. Here we show that MiDAC is important for chromosome alignment during mitosis in cancer cell lines. Mice lacking the MiDAC proteins, DNTTIP1 or MIDEAS, die with identical phenotypes during late embryogenesis due to perturbations in gene expression that result in heart malformation and haematopoietic failure. This suggests that MiDAC has an essential and unique function that cannot be compensated by other HDAC complexes. Consistent with this, the cryoEM structure of MiDAC reveals a unique and distinctive mode of assembly. Four copies of HDAC1 are positioned at the periphery with outward-facing active sites suggesting that the complex may target multiple nucleosomes implying a processive deacetylase function.

[1] Leicester Institute of Structural and Chemical Biology, University of Leicester, Leicester LE1 7RH, UK. [2] Department of Molecular and Cell Biology, University of Leicester, Leicester LE1 7RH, UK. [3] School of Life Sciences, University of Warwick, Coventry CV4 7AL, UK. [4] Division of Molecular and Cellular Function, University of Manchester, Manchester M13 9PL, UK. [5] Present address: Institute of Clinical Sciences, Faculty of Medicine, Imperial College, Hammersmith Hospital Campus, Du Cane Road, London W12 0HS, UK. [6] Present address: AstraZeneca, Milstein Building, Granta Park, Cambridge CB21 6GH, UK. [7] Present address: MRC London Institute of Medical Sciences, Hammersmith Hospital Campus, Du Cane Road, London W12 0NN, UK. [8] Present address: Department of Genetics, University of Cambridge, Downing Street, Cambridge CB2 3EH, UK. [9] These authors contributed equally: Robert E. Turnbull, Louise Fairall, Almutasem Saleh, Emma Kelsall. ✉email: smc57@leicester.ac.uk; john.schwabe@leicester.ac.uk

Class I histone deacetylases (HDACs 1–3) are essential regulators of gene expression. The enzymes are recruited to chromatin as part of large multi-protein complexes that control the acetylation state of histones and other chromatin-associated factors. These are high stoichiometry acetylation sites whose regulation controls gene transcription[1].

In addition to the HDAC catalytic subunit, these complexes contain scaffold proteins that mediate interaction with transcription factors and chromatin, thereby determining the specificity of these complexes. Importantly, class I HDACs have enhanced catalytic activity when assembled into their cognate complexes and their activity is further increased by binding higher order inositol phosphates[2,3]. Inhibitors of class I HDACs have been used to target a number of diseases, including HIV, Alzheimer's disease and cancer[4–7].

The best-studied complexes include the NuRD, Sin3 and CoREST complexes which contain HDAC1/2, and the SMRT/NCoR complex that contains HDAC3[8–11]. The MiDAC complex, the focus of this study, is relatively poorly understood. The complex contains three proteins: HDAC1/2, MIDEAS (aka. ELMSAN1, c14orf43) and DNTTIP1 (aka. TDIF1). It was first identified through chemoproteomic approaches. The complex was recruited to an HDAC-inhibitor-bound resin in cells stalled in mitosis by nocodazole treatment—hence the name mitotic deacetylase complex (MiDAC)[12]. Subsequent proteomic studies supported the identification of the MiDAC complex[13–16]. The complex is present in diverse species such as nematodes and jelly fish, suggesting that it plays an important role that has been conserved through evolution. Further evidence that the MiDAC complex has a role in the cell cycle comes from the observation that MiDAC components are associated with, and are substrates of, the CyclinA2/CDK2 complex[17].

Consistent with a role in cell proliferation, components of the MiDAC complex have been implicated in a number of human cancers. DNTTIP1 has a critical role in oral cancer and is proposed to be oncogenic in non-small cell lung cancers[18,19]. A number of large-scale cancer genome studies have associated downregulation of MIDEAS with cutaneous melanoma; MIDEAS as a mutational hotspot and potentially a rare tumour gene[20–22].

The evolutionary conservation of MiDAC, together with apparent roles in the cell cycle and in several cancers suggests that MiDAC is an important deacetylase complex. In this study, to gain a better understanding of MiDAC's biological role, we explore the effects of depleting MiDAC in cells and mice. Cell-based assays show that the endogenous complex is present throughout the cell cycle, but that siRNA knockdown of either DNTTIP1 or MIDEAS results in increased metaphase chromosome misalignment. Homozygous mice embryos lacking either MIDEAS or DNTTIP1 die after day e16.5 with identical phenotypes. The embryos are severely anaemic and have a clear malformation of the heart. Gene expression analyses in fibroblasts derived from the homozygous embryos show a significant overlap in the gene sets whose expression is perturbed, many of which are involved in developmental pathways.

To understand how the MiDAC complex is assembled and how it is distinct from other HDAC complexes, we use cryo-EM to determine the structure of the complex. Unexpectedly, the ELM2 domain of MIDEAS mediates dimerisation through interaction with DNTTIP1. The MIDEAS-SANT domain mediates tetramerisation of the complex. The overall assembly of the complex is critically dependent on the interactions between DNTTIP1 and MIDEAS. The tetrameric architecture resembles a three-dimensional X-shape with the HDAC catalytic sites at the four extremities of the complex, suggesting that the complex is able to simultaneously target multiple nucleosomes and that it might therefore be a highly processive deacetylase complex. Both

the knockout mice and the cryo-EM structure are consistent with co-dependency of DNTTIP1 and MIDEAS for both the structure and function of the MiDAC complex. These data provide an important step change in our understanding of the mammalian MiDAC complex.

## Results

**MIDEAS, DNTTIP1 and HDAC1 co-localise through the cell cycle.** To investigate the sub-cellular distribution of MIDEAS, DNTTIP1 and HDAC1, we used immunofluorescence in human U2OS cells. Antibodies were selected based on the epitope being both unique and characterised for specificity using GFP-MIDEAS and mCherry-DNTTIP1 (see below and Supplementary Fig. 1a–d).

HDAC1, MIDEAS and DNTTIP1 are located predominantly in the soluble nuclear fraction in U2OS cells (Fig. 1a). Moreover, immunoprecipitation with antibodies against MIDEAS or DNTTIP1 showed HDAC activity significantly greater than in a control IP (Fig. 1b). This activity was blocked by the HDAC inhibitor SAHA and potentiated by inositol hexakisphosphate (InsP6) (Fig. 1c).

Previously, Bantscheff et al.[12] reported that an HDAC-inhibitor matrix preferentially pulls down DNTTIP1 in cells blocked in the mitosis by nocodazole. To explore whether this is explained by variation in expression through the cell cycle or changes in localisation, we used immunofluorescence microscopy with HDAC1, MIDEAS and DNTTIP1 antibodies to visualise the endogenous proteins at multiple cell-cycle stages (Fig. 1d). HDAC1, MIDEAS and DNTTIP1 showed nuclear localisation throughout interphase, but started to become excluded from chromatin as chromosomes condensed during early mitosis (prophase). The proteins were then recruited back into nuclei as they re-formed and the chromatin decondensed during late mitosis (telophase) (Fig. 1d).

To explore this further, we used CRISPR/Cas9-mediated gene editing to introduce a FLAG epitope onto the C-terminus of the endogenous MIDEAS protein in mouse embryonic stem (ES) cells (Supplementary Fig. 1e, f). Western blotting confirmed that both MIDEAS and DNTTIP1 are present at all stages of the cell cycle (Fig. 1e; Supplementary Fig. 2a) and that MIDEAS-FLAG pulled-down DNTTIP1 throughout the cell cycle (Fig. 1f; Supplementary Fig. 2b). Furthermore, in U2OS cells, antibodies against MIDEAS and DNTTIP1 were able to co-immunoprecipitate the other protein as well as both HDACs 1 and 2 in asynchronous cells and cells in both G1/S and M phase (Fig. 1g; Supplementary Fig. 2c). The HDAC activity of immunoprecipitated complexes from G1/S cells showed slightly higher activity than either asynchronous or mitotic cells (Fig. 1h; Supplementary Fig. 2c).

**MIDEAS and DNTTIP1 protein expression is co-dependent.** To investigate the role of the MiDAC complex in U2OS cells, we obtained two independent siRNAs against both MIDEAS and DNTTIP1 (two against each transcript). Both MIDEAS siRNAs significantly reduced MIDEAS mRNA expression relative to control. Similarly, siRNAs directed against DNTTIP1 effectively abolished DNTTIP1 mRNA expression (Supplementary Fig. 3a, b). Each siRNA was specific to its target, and there was no effect on the mRNA expression level of the other components of the complex (Supplementary Fig. 3a–c). Western blotting using the soluble nuclear fraction from cells treated with siRNAs against both DNTTIP1 and MIDEAS showed significantly reduced protein levels for both MIDEAS and DNTTIP1, but not HDAC1 (Fig. 2a; Supplementary Fig. 3d). These results suggest that protein expression and stability of MIDEAS and DNTTIP1 are mutually co-dependent whilst that of HDAC1, probably due to its

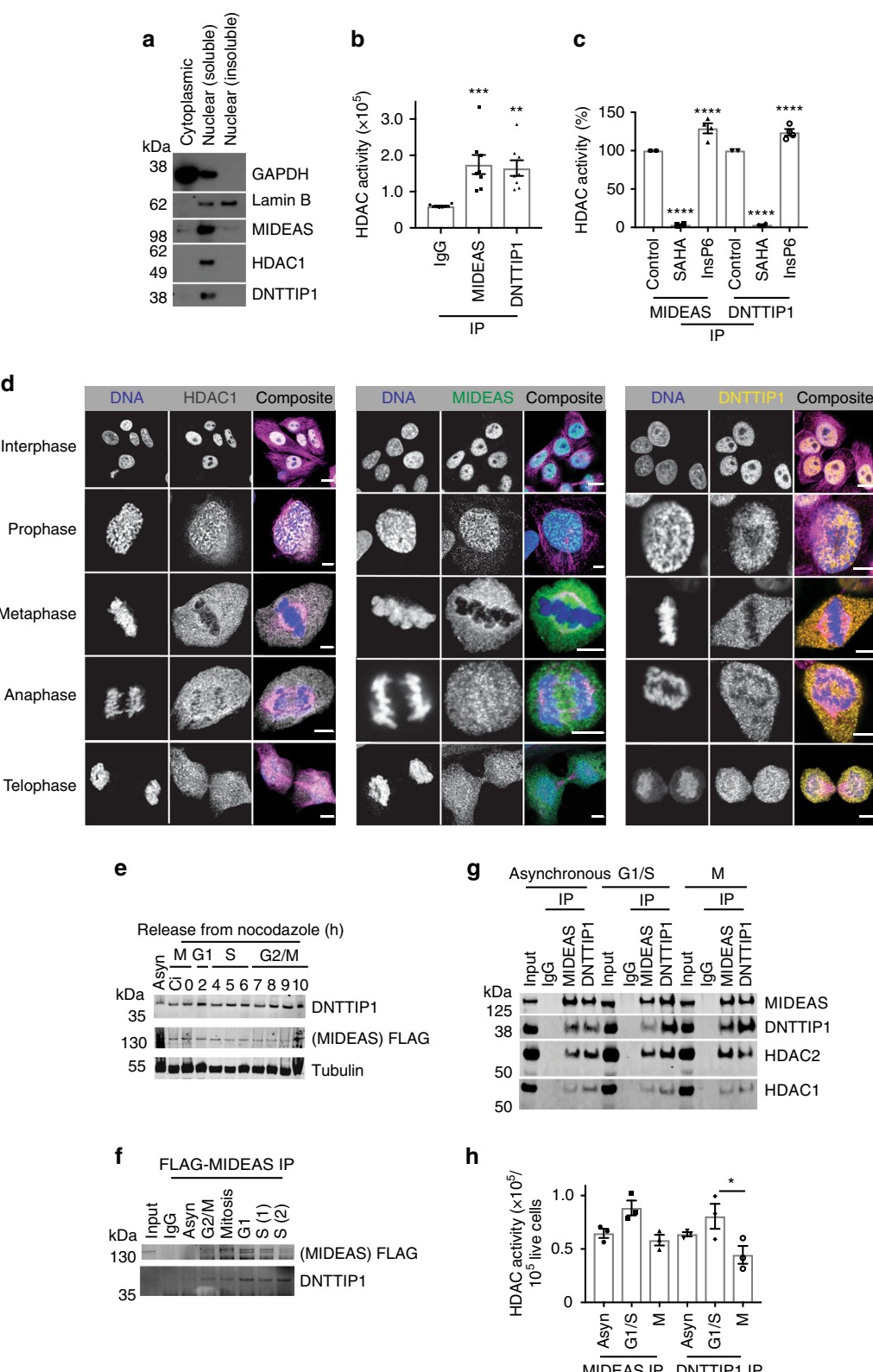

participation in other complexes, is unaffected. This observation that DNTTIP1 and MIDEAS protein levels are co-dependent in cells fits well with the architecture of the MiDAC complex (see below).

**MiDAC depletion increases mitotic chromosome misalignment.** siRNA knockdown of either MIDEAS or DNTTIP1 in

asynchronous U2OS cells, resulted in a significant increase in misaligned chromosomes during metaphase. Figure 2b shows representative images of control and siRNA-treated metaphase cells. The misalignment in cells with reduced MIDEAS and DNTTIP1 correlates with aberrant DNA and CENPA staining away from the metaphase plate. Quantification of this observation showed that both MIDEAS and DNTTIP1 siRNA's increased

**Fig. 1 Nuclear localisation and HDAC activity of the endogenous MiDAC complex. a** Cytoplasmic and nuclear proteins were prepared from U2OS cells before western blotting using antibodies against MIDEAS, DNTTIP1 and HDAC1. Antibodies against GAPDH and LaminB were used, respectively, as cytoplasmic and nuclear housekeeping controls. **b** Fluorescent-based HDAC activity assay of IgG control, MIDEAS and DNTTIP1 immunoprecipitates from U2OS nuclear lysates (mean ± s.e.m, $n = 8$ independent experiments, ***$P = 0.001$, **$P = 0.0012$, one-way ANOVA with Holm–Sidak post hoc test). **c** Fluorescent- based HDAC activity assay of MIDEAS and DNTTIP1 immunoprecipitates from nuclear lysates that had been pre-incubated with 100 μM InsP6 or 5 μM SAHA for 30 min. IgG controls were performed in parallel and used to subtract background activity before normalisation to untreated complex HDAC activity (mean ± s.e.m, $n = 4$ independent experiments, ****$P < 0.0001$, one-way ANOVA with Holm–Sidak post hoc test). **d** Confocal images of PFA fixed U2OS cells dual stained with either HDAC1 (grey), MIDEAS (green) or DNTTIP1 (yellow) antibodies and anti-α-tubulin (magenta). DNA (blue) was visualised with Hoechst 33258 (scale bars: interphase, 10 μm; mitotic cells, 5 μm). **e** Western blot for DNTTIP1, MIDEAS-FLAG and Tubulin using lysates from MIDEAS-FLAG CRISPR mouse ES cells that had been incubated for 14 h with 10 μM CDKi RO-3306 (Ci), followed by a further 2-h incubation with nocodazole (30 ng/ml). Cells were harvested at various time points after nocodazole release. **f** Western blot for MIDEAS-FLAG and DNTTIP1 from cell-cycle synchronised cells following co-immunoprecipitation (Co-IP) using a anti-FLAG antibody. Cells synchronised as in panel **e**. **g** Western blot for MIDEAS, DNTTIP1, HDAC2 and HDAC1 from co-IP's using Rb anti-MIDEAS and Rb anti-DNTTIP1 antibodies. U2OS cells were blocked for 16 h in the G1/S with aphidicolin (1.6 μg/ml) or M with nocodazole (3.3 μg/ml). A mock IP was also carried out with rabbit IgG as a control. Antibodies were cross-linked to beads prior to IP. **h** Fluorescent-based HDAC activity assay of MIDEAS and DNTTIP1 immunoprecipitates from U2OS cells blocked in the G1/S or M as in panel **g**. Results are normalised to cell count after detachment from plates (mean ± s.e.m, $n = 3$ independent experiments, *$P = 0.022$, one-way ANOVA with Holm–Sidak post hoc test). Source data are provided as a Source data file.

misalignment by 1.5-fold and twofold, respectively, compared with control transfections (Fig. 2c). To check this result was not specific to U2OS cells, we also performed siRNA-mediated knockdown of MIDEAS in HeLa cells and observed a similar increase in chromosome misalignment (Fig. 2d).

To confirm that the misalignment phenotype is due to loss of the MiDAC complex and not off-target effects of the siRNAs, we created a stable U2OS cell line with a doxycycline-inducible siRNA-resistant FLAG-DNTTIP1. Western blotting confirmed that the FLAG epitope could only be detected in the presence of doxycycline (Fig. 2e). Addition of DNTTIP1 siRNA decreased endogenous DNTTIP1 protein levels regardless of whether doxycycline was present or not, but had no effect on the expression of the siRNA-resistant FLAG-DNTTIP1 (Fig. 2e; Supplementary Fig. 3e–g). Both siRNAs against DNTTIP1 increased chromosome misalignment in the absence of doxycycline as seen previously. As expected, induction of FLAG-DNTTIP1 with doxycycline reduced misalignment to a level comparable with control transfections (Fig. 2f).

**MiDAC plays an essential role in late embryogenesis**. To explore the role of the MiDAC complex in vivo, we used CRISPR-Cas9-mediated gene editing to generate KO alleles for both *Mideas* and *Dnttip1*. crRNAs targeting exon 2 of either *Mideas* and *Dnttip1* were injected into single-cell zygotes to generate 10-bp and 11-bp deletions, respectively. These modified alleles produce a premature stop codon within the open-reading frames of both genes leading to a constitutive KO phenotype (Supplementary Fig. 4). Heterozygous mice were healthy and fertile and so were inter-crossed to generate homozygous animals. Genotyping the resulting litters revealed a complete absence of viable homozygous pups from both MIDEAS-del1 and DNTTIP1-del1 heterozygous crosses, indicating an essential role for the MiDAC complex during embryogenesis (Supplementary Table 1).

To investigate the stage at which the homozygous embryos die, we performed a series of timed matings. We observed homozygous embryos at days e13.5, e14.5, e15.5 and e16.5. Strikingly, the homozygous embryos are readily identified through their pale colour and somewhat smaller size than the wild-type or heterozygous embryos (Fig. 3a; Supplementary Fig. 5a, b).

Stained sections of the homozygous embryos showed a remarkable lack of red blood cells in the heart and vessels throughout the body, as well as morphological differences in the heart itself (Fig. 3b; Supplementary Fig. 5c, d). The heart is smaller and malformed with an enlarged pericardium. The lack of blood would suggest either a failure of haematopoiesis; substantial

vascular leakage or perhaps abnormally rapid turnover of blood cells. The spleen and liver are the main sites of haematopoiesis at this stage of development. However, these tissues in the mutant animals appear to be morphologically normal with comparable numbers of hematopoietic precursors (Supplementary Fig. 5d).

**MiDAC knockout perturbs multiple gene-regulatory networks**. To determine the gene-regulatory consequences of the MIDEAS and DNTTIP1 deletions, we prepared mouse embryonic fibroblasts (MEFs) from day e13.5 embryos. Wild-type (+/+) and homozygous (−/−) lines were established from the MIDEAS-del1 and DNTTIP1-del1 embryos. Successful gene deletions were confirmed by genotyping and loss of HDAC activity in a co-IP assay or loss of protein by western blot analysis (Supplementary Fig. 6a–d). Interestingly, for both gene deletions, there was no apparent defect in cell proliferation or cell-cycle progression, consistent with the observed late developmental failure (Supplementary Fig. 6e–h). Importantly, however, we observed a significant increase in chromosome misalignment during metaphase in both MIDEAS$^{−/−}$ and DNTTIP1-del1$^{−/−}$ MEFs (Fig. 3c).

To further investigate the possible mechanisms for increased chromosome misalignment, heart malformation and likely haematopoietic failure, the total RNA was isolated from wild-type, MIDEAS$^{−/−}$ and DNTTIP1$^{−/−}$ MEF lines and used for RNAseq analysis. In total, 468 differentially expressed transcripts were common to both the MIDEAS and DNTTIP1 knockouts (Fig. 3d) with more transcripts showing increased than decreased expression (Fig. 3e). The changes in the common transcripts showed a clear correlation ($R^2 = 0.91$) in both direction and magnitude of altered expression in MIDEAS$^{−/−}$ and DNTTIP1$^{−/−}$ cells, again confirming the co-dependence of MIDEAS and DNTTIP1 (Fig. 3f).

DAVID[23] and Panther[24] pathway analyses indicate that the set of overlapping, upregulated genes from the knockout cell lines are enriched in genes involved in developmental pathways, in particular Wnt signalling, Cadherin signalling and axon guidance. Other pathways affected include FGF and EGF signalling and angiogenesis (Fig. 3g; Supplementary Fig. 5e and Supplementary Table 2). Interestingly, analysis based on tissue specificity suggests that many of these genes are involved in brain development (Fig. 3h). Normally, these genes would not be expected to be expressed in fibroblasts. It is likely that one of the roles of the MiDAC complex is to repress expression of these genes.

Gene ontology analysis also identified some genes involved in cell-cycle regulation (Supplementary Table 3), and gene set enrichment analysis (GSEA; Broad Institute[25]) showed that there

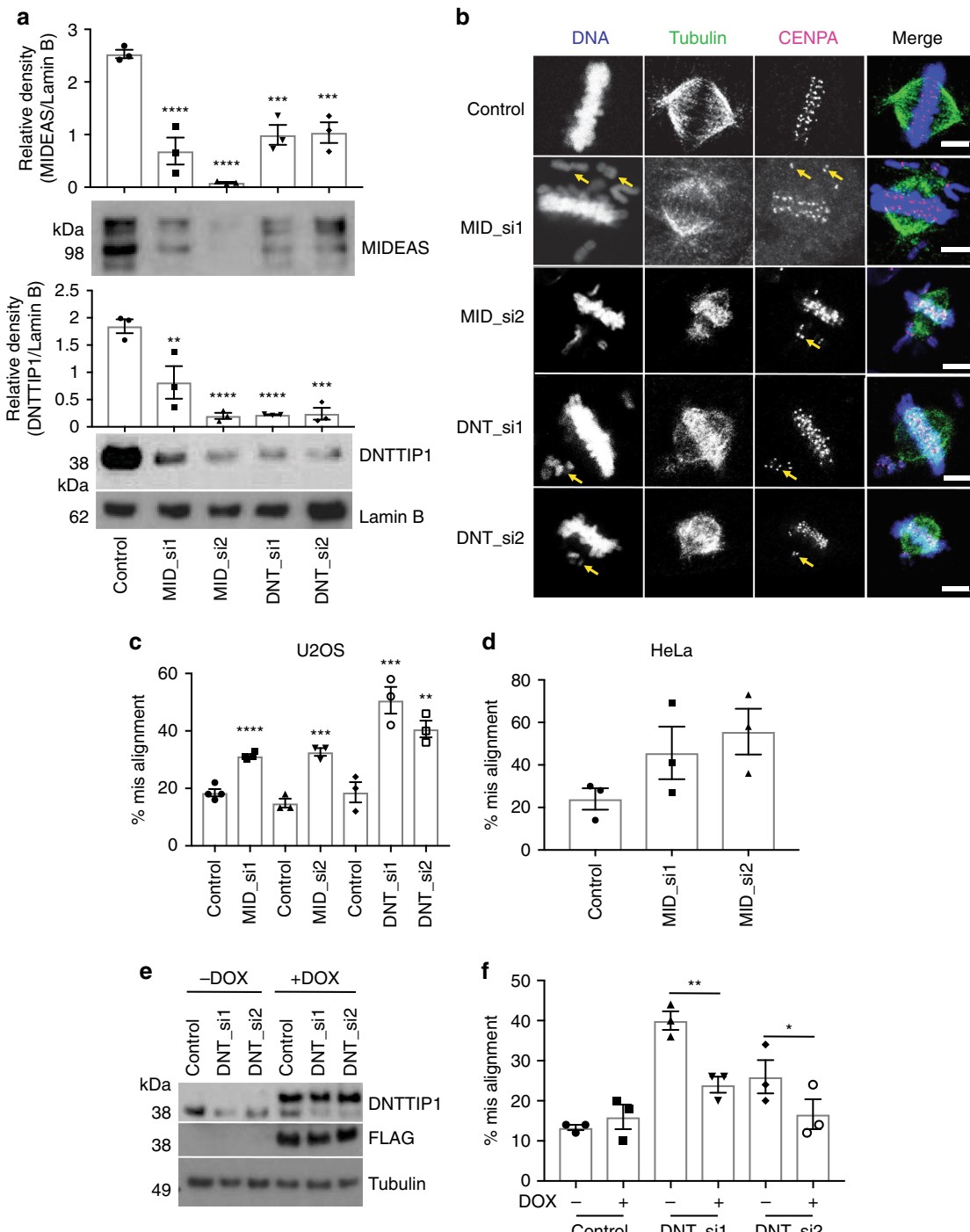

**Fig. 2 MiDAC protein co-dependency and role in chromosome alignment. a** Western blot for MIDEAS (top) and DNTTIP1 (bottom) following treatment of U2OS cells with siRNA for 72 h targeting MIDEAS or DNTTIP1. Blots shown are representative. LaminB was detected in parallel from the same lysate sample, and was used as a reference for normalisation (mean ± s.e.m., n = 3 independent experiments, **P = 0.0029, ***P < 0.001 (P = 0.0003 for control versus DNT_si1, 0.0004 for control versus DNT_si2), ****P < 0.0001, one-way ANOVA with Dunnett's post hoc test). **b** Confocal images representative of normal and misaligned chromosomes. U2OS cells were treated with siRNA as in (**a**) before fixation with PFA and staining for α-tubulin (green), CENPA (magenta). DNA (blue) was visualised with Hoechst 33258 (scale bars: 5 μm). **c** Quantification of metaphase cells with misaligned chromosomes in U2OS cells after siRNA treatment shown as percent misaligned (mean ± s.e.m, n = 3 MID_si2, DNT_si1, DNT_si2, n = 4 MID_si1, (50 metaphase cells counted over 3/4 independent experiments), ***P < 0.001 (P = 0.0008 for control versus MID_si1, 0.0003 for control versus MID_si2), ****P < 0.0001, one-way ANOVA with Holm–Sidak post hoc test). **d** Quantification of metaphase cells with misaligned chromosomes in HeLa cells after MIDEAS siRNA treatment shown as percent misaligned (mean ± s.e.m., n = 3 (50 metaphase cells counted over 3 independent experiments). **e** Western blot using nuclear proteins isolated from stable, DOX inducible, siRNA-resistant FLAG-DNTTIP1 U2OS cell line showing DOX induces FLAG-DNTTIP1. **f** Quantification of metaphase cells with misaligned chromosomes in U2OS cells after siRNA treatment and induction of siRNA-resistant FLAG-DNTTIP1 with DOX (mean ± s.e.m., n = 3 (50 metaphase cells counted over three individual experiments), *P = 0.0416, **P = 0.0041, one-way ANOVA with Holm–Sidak post hoc test). Source data are provided as a Source data file.

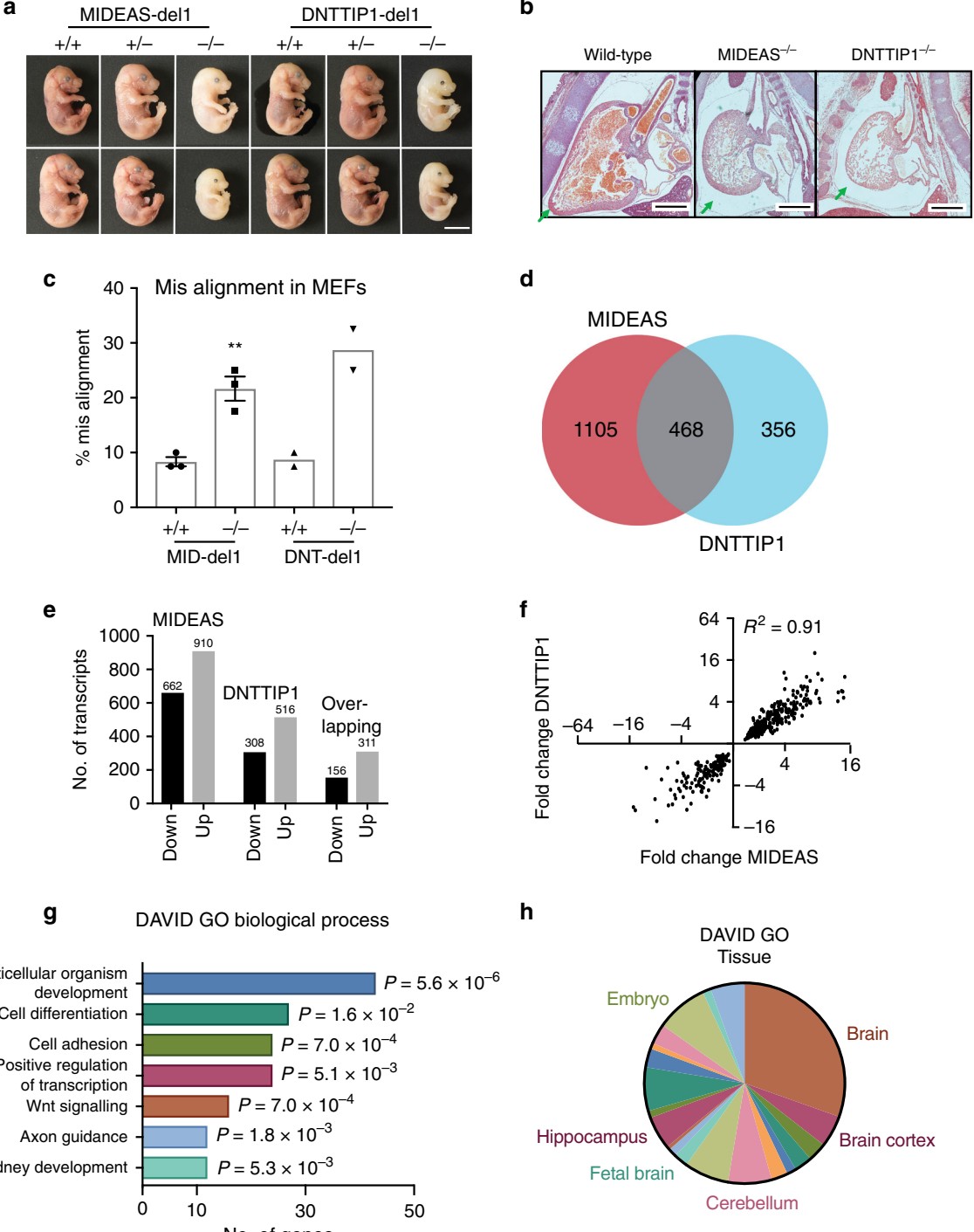

**Fig. 3 Analysis of mice embryos and MEF's lacking MIDEAS or DNTTIP1. a** Images of wild-type, heterozygous and homozygous MIDEAS-del1 and DNTTIP1-del1 embryos isolated at e16.5 (scale: 5 mm). **b** Images of sections from e16.5 wild-type, MIDEAS$^{-/-}$ and DNTTIP1$^{-/-}$ embryos demonstrating absence of erythrocytes in the heart, enlarged pericardium and deformed ventricle morphology in the knockouts compared with wild-type (green arrows) (scale: 500 μm) (representative images from $n = 2$ biologically independent animals). **c** Quantification of metaphase cells with misaligned chromosomes in mouse embryonic fibroblasts (MEFs) from e13.5 MIDEAS-del1 (MID-del1) or DNTTIP1-del1 (DNT-del1) wild-type and homozygous embryos (mean ± s.e.m., $n = 3$ MID-del1, $n = 2$ DNT-del1 (40 metaphase cells were counted over 3/2 individual experiments), $**P = 0.0048$, two-tailed unpaired Student's $t$ test). **d** Venn diagram depicting the number of overlapping genes identified as differentially expressed in MIDEAS and DNTTIP1 knockout MEFs. Differential expression was based on a $P$-value of <0.1 as calculated using DESEQ2 in R. **e** The number of down- and upregulated genes identified as differentially expressed in MIDEAS and DNTTIP1 knockout MEFs and in the overlapping gene set list from panel **d**. Differential expression calculated from $n = 3$ (MIDEAS knockout MEFs) and $n = 2$ (DNTTIP1 knockout MEFs) independent MEF lines from e13.5 embryos, $P$-value of <0.1 as calculated using DESEQ2 in R. **f** Plot of the 468 overlapping perturbed gene set showing fold change for homozygous knockout MIDEAS ($x$-axis) and DNTTIP1 ($y$-axis) genes. ($R^2$ calculated using Pearson correlation). **g** DAVID gene ontology (GO) biological processes analysis using the overlapping, upregulated gene list. List shows changes in biological process with more than 12 genes upregulated from the overlapping gene list and a Benjamini post hoc test with $P < 0.05$. **h** DAVID GO tissue association of the overlapping, upregulated gene list. Source data are provided as a Source data file.

was a reduction in the expression of transcripts associated with mitotic spindle organisation (Supplementary Fig. 5f and Supplementary Table 4). This provides a possible explanation for the increased misalignment seen in knockout MEFs and siRNA knockdowns of MIDEAS and DNTTIP1.

**The MiDAC dimer complex reveals a distinct mode of assembly.** The finding that MiDAC has a unique functional role that cannot be compensated for by other HDAC complexes prompted us to ask whether the assembly of the complex might also be unique and so sought to determine the structure. We have shown previously that a complex with full-length HDAC1, full-length DNTTIP1 and the ELM2-SANT domain from MIDEAS forms a stable 450-kDa tetramer in solution—a size suitable for structure determination using cryo-EM[26] (Fig. 4a). We explored a number of conditions to prepare samples of this complex for cryo-EM. Maps obtained from a grafix cross-linked tetrameric complex were limited to only around 15-Å resolution. Gentler cross-linking resulted in a sample that was partly

dissociated during grid preparation, and both dimers and tetramers were observed. By careful masking we were able to obtain maps of the dimer at ~6-Å resolution, but only 23-Å resolution for the tetramer (Supplementary Fig. 7). It became apparent that the DNA-binding domain of DNTTIP1 could not be observed in maps of either the dimer or tetramer, suggesting that it is flexibly attached. To overcome this issue, we prepared a smaller complex lacking the DNA-binding domain of DNTTIP1 (Fig. 4b). We also included InsP6 and an HDAC inhibitor (SAHA) since both have previously been shown to stabilise the HDAC3:SMRT complex[27]. This smaller complex was cross-linked on ice and frozen onto UltrAuFoil EM grids. We collected 2752 micrographs using a Volta Phase Plate on a Titan Krios G3 with a Falcon III camera (Fig. 4c). As before, we observed a mixture of dimeric and tetrameric particles and were able to obtain detailed class averages (Fig. 4d, e). In all, 126,484 particles were included in the final ~4-Å map of the dimeric complex with C2 symmetry applied (Fig. 5a; Supplementary Figs. 8 and 9a–c and Supplementary Table 5). For the tetramer,

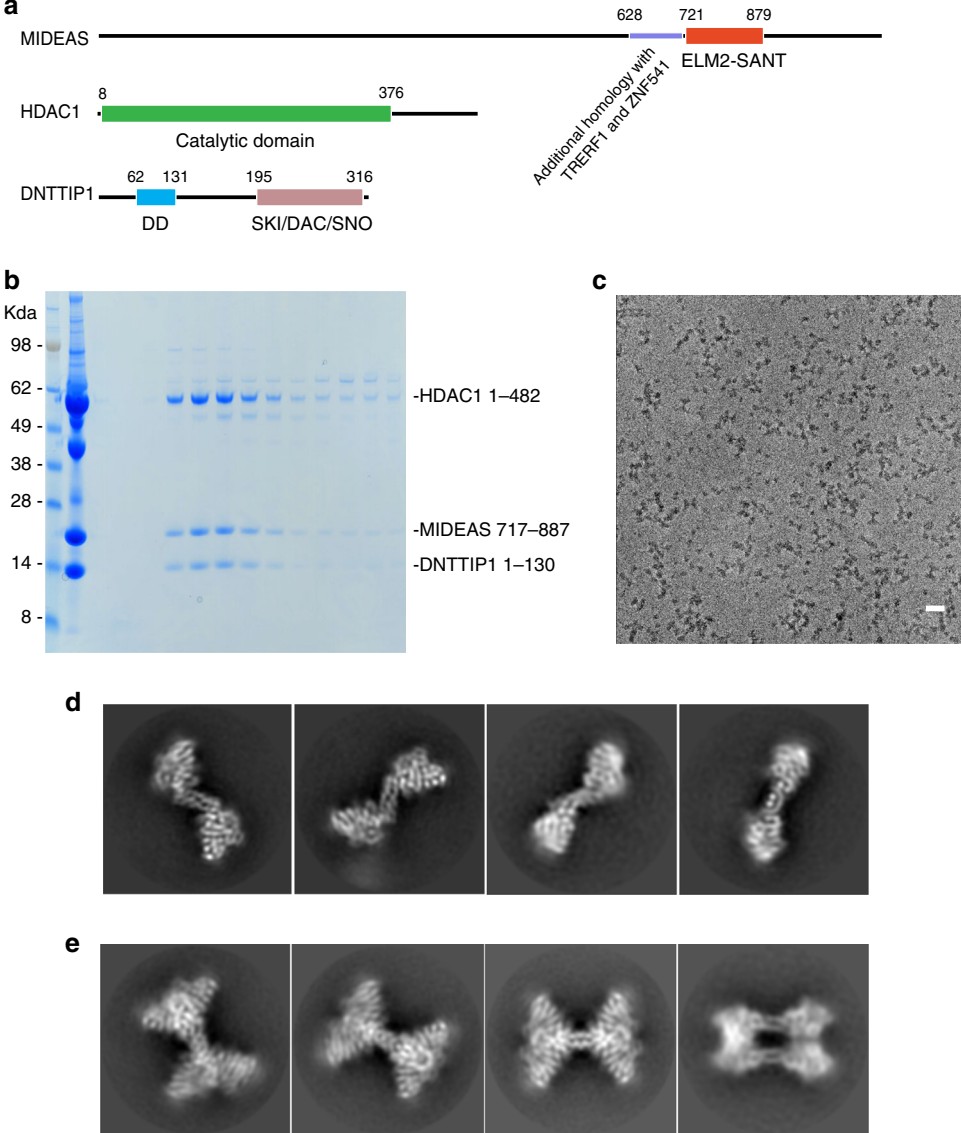

**Fig. 4 Cryo-electron microscopy of the MiDAC complex. a** Schematic of the domain structures of MIDEAS, HDAC1 and DNTTIP1: components of the MiDAC complex. **b** SDS-PAGE of the gel-filtration purification of the MiDAC complex on a Superdex-S200 column. **c** Section of an electron micrograph of the MiDAC complex. Scale bar: 20 nm. **d** 2D class averages of the dimer complex from Relion3. **e** 2D class averages of the tetramer complex from Relion3.

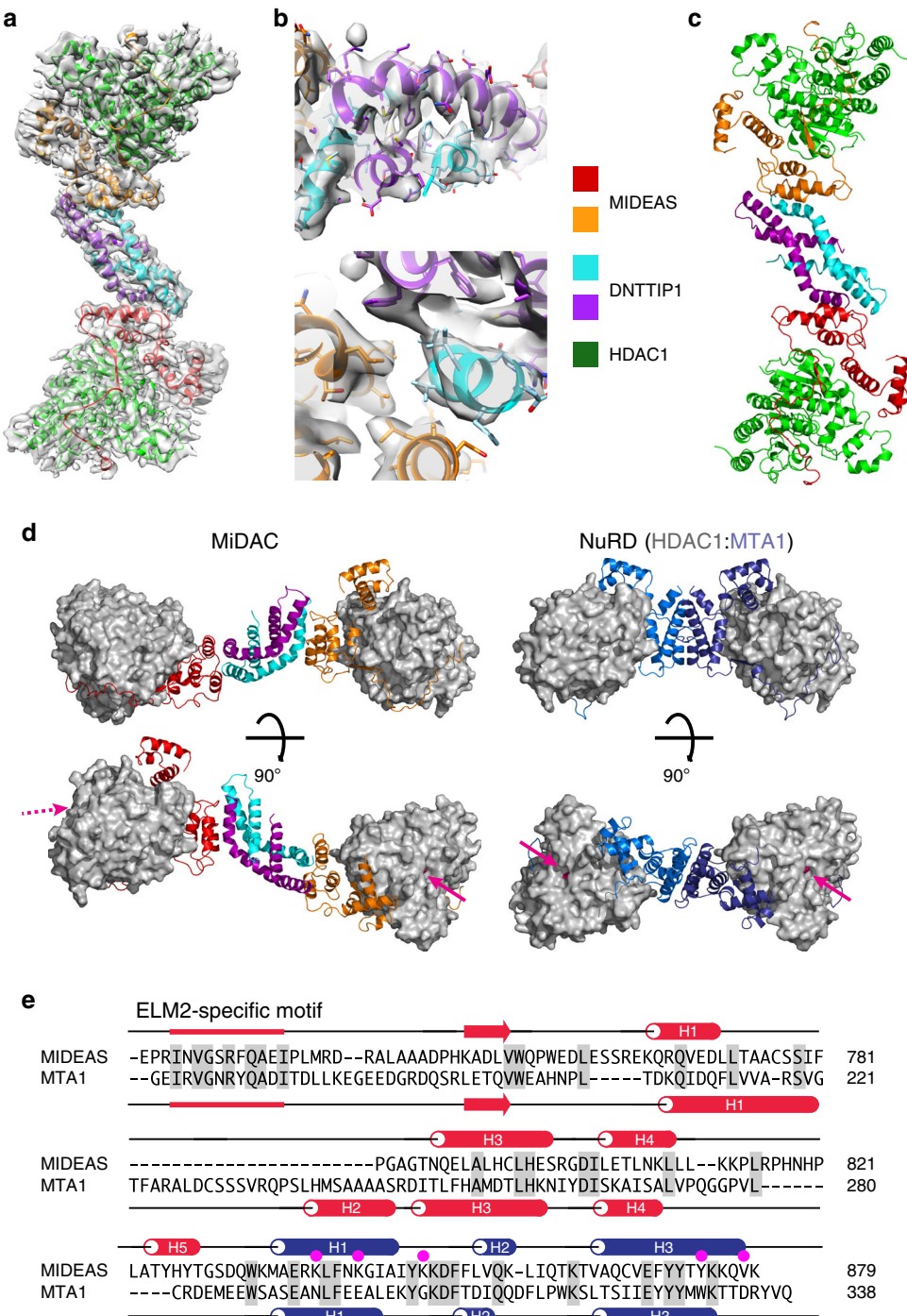

**Fig. 5 The structure of the dimeric MiDAC complex. a** Post-processed cryo-EM map of the dimeric complex from Relion3 with protein chains shown as cartoons. Map contour level is 0.024—Chimera. **b** Closeup of the dimerisation domain of DNTTIP1 and a closeup of the dimerisation domain of DNTTIP1 interacting with MIDEAS. Map contour level is 0.024—Chimera. **c** Cartoon representation of the dimeric MiDAC complex. **d** Comparison of the MiDAC complex with the NuRD complex. The active sites are shown with magenta arrows. The dotted magenta arrow indicates that the active site is at the back. **e** Structural alignment of MIDEAS with MTA1. The alpha-helices are shown as cylinders, and the beta-strand as an arrow. The ELM2 domain is shown in red, and the SANT domain in blue. Identical residues are shaded grey, and residues at the tetramer interface are indicated by magenta spots.

63,222 particles were included in the final 4.5-Å map with D2 symmetry applied (Supplementary Figs. 8 and 9d–f and Supplementary Table 5). Both the dimer and tetramer grids suffer from some preferential orientation and map anisotropy (Supplementary Fig. 9).

The known crystal structures of HDAC1[3] and the dimerisation domain of DNTTIP1[26] could easily be recognised in the maps and were docked accordingly. The crystal structures of HDAC1

and the dimerisation domain of DNTTIP1 fitted extremely well in the EM maps and no attempt was made to rebuild the side chains of these proteins. Models for the MIDEAS co-repressor protein were obtained using iTASSER and PHYRE based on the MTA1 structure from the HDAC1:MTA1 crystal structure[3,28,29]. MIDEAS was rebuilt into the map taking into account secondary structure predictions. Rigid body refinement of all three proteins and one round of simulated annealing were used to optimise the

fit into the map and to remove obvious clashes. Density for the InsP6, but not the SAHA, was clearly visible in the map (Supplementary Fig. 10a).

The structure of the dimeric complex has an overall S shape (Fig. 5a). It reveals that the ELM2 domain from MIDEAS does not mediate dimerisation. This was unexpected since the homologous ELM2 domain from MTA1 directly mediates dimerisation in the NuRD complex[3]. Instead, the MIDEAS ELM2 domain interacts with the DNTTIP1 dimerisation domain which mediates the dimeric assembly (Fig. 5b, c). The consequence of this is that the HDAC1 catalytic subunits are arranged very differently, located much farther apart, with the HDAC1 active sites on opposite ends of the dimer (Fig. 5d). This is a complete contrast with the NuRD complex in which MTA1 dimerisation results in the HDAC1 active sites being positioned on the same face of the complex (Fig. 5d).

A structure-based alignment of the ELM2-SANT domains from MIDEAS and MTA1 reveals the differences that give rise to the distinct architectures of the two complexes (Fig. 5e). The first helix of MTA1-ELM2 is significantly longer than the equivalent helix in MIDEAS. The shorter helix in MIDEAS mediates a tight interaction with a non-polar groove between the end of the long helices of the DNTTIP1 dimer (Supplementary Fig. 10b). Other differences are that MIDEAS lacks helix 2 of the ELM2 domain which contributes to the homodimer interface in MTA1 (Fig. 5e). MIDEAS also has an extra helix before the linker leading to the SANT domain. The SANT domains of MIDEAS and MTA1 are very similar in structure to each other and to the SANT domain of the SMRT protein that binds to HDAC3[2] (Supplementary Fig. 10c). One of the regions of MIDEAS most similar to other ELM2-SANT domain co-repressor proteins is the ELM2-specific motif which binds in an extended conformation in a conserved groove on HDAC1 (Supplementary Fig. 10c). This interaction appears to be important for tethering the co-repressor such that it wraps around the catalytic domain of the HDAC.

**MiDAC tetramerization is mediated by both MIDEAS and DNTTIP1.** The final map of the tetrameric particles was refined using D2 symmetry to ~4.5 Å (Supplementary Figs. 8 and 9). The DNTTIP1 dimerisation domain and the core of HDAC1 are somewhat more ordered than other parts of the structure (Supplementary Fig. 9e). The tetrameric complex clearly contains a dimer of the dimeric complexes with no obvious structural perturbations (Fig. 6a, b). The arrangement is such that the DNTTIP1 dimerisation domains are symmetrically juxtaposed, but do not directly interact (Fig. 6b; Supplementary Fig. 11). The two S-shaped dimers are rotated 180° with respect to each other with the four HDAC subunits positioned at the periphery of the tetramer, which results in a striking three-dimensional X-shaped complex (Fig. 6b; Supplementary Fig. 11). This three-dimensional X-shape of the tetrameric complex positions the active sites of the four HDACs at the extremities of the complex (Fig. 6c).

The tetramerisation interface between the two dimers is mediated by interactions between the SANT domains (Fig. 6c). Residues Lys849, Tyr874 and Val878 contribute to the interface between the two neighbouring SANT domains (Supplementary Fig. 10d). These residues are not conserved in MTA1 (Fig. 5e). Interestingly, Lys839 and Lys843 appear to be in a position to interact with the inositol phosphate that is bound to the neighbouring dimer, although there is no apparent density for the side chains (Supplementary Fig. 10d, e). This may explain why it was only possible to obtain a high-resolution map of the tetramer in the presence of inositol phosphate (7.7 Å vs 4.5 Å). A further obvious effect of the InsP6 was to significantly improve

the local resolution of the core of the HDAC1, consistent with the proposed mechanism of activation[27] (Supplementary Fig. 10f).

In addition to the interactions between the SANT domains it appears that the residues N-terminal to the DNTTIP1 dimerisation domain are crossing over to interact with the neighbouring tetramer. Unfortunately, the quality of the map in this area does not enable de novo model building (Supplementary Fig. 10g). The importance of the N-terminal region of DNTTIP1 for tetramerisation is supported by biochemical experiments which suggest that a complex lacking residues 1–49 of DNTTIP1 is no longer tetrameric based on a size-exclusion column (Supplementary Fig. 12).

**Position of the DNTTIP1 chromatin-binding domain.** We also analysed cryo-EM grids of complexes containing full-length DNTTIP1, including the DNA-binding domain (Supplementary Figs. 7 and 12e–g). However, we were only able to achieve low-resolution (~23 Å) maps with this complex. The overall shape of the complex (with no symmetry applied) is the same as the complex without the DBD, but there is additional, ill-defined density, presumably from the DBD, around the middle of the complex (Fig. 6d). This suggests that the DBDs are flexibly linked to the tetrameric core.

Interestingly, when we masked the tetrameric core of the complex and applied D2 symmetry, we were only able to improve the resolution to 16 Å (Fig. 6e). It is not clear whether the flexible DBDs are introducing genuine structural heterogeneity into the core of the complex or whether the DBDs prevent the processing algorithms from generating an accurate alignment of the particles.

Flexible attachment of DBDs to the rest of the protein is a common feature of DNA-binding proteins and is reminiscent of the DNA-binding domain of the telomere-binding protein TRF1 which can bind to telomeric sequences with widely variable spacings[30]. Flexibility would allow the complex to interact with chromatin in different conformations.

**Discussion**

HDACs 1 and 2 are assembled into multiple distinct complexes. Specific proteins in each of these complexes target them to particular chromatin loci, depending on the role of the particular complex. Many of the complexes use a conserved ELM2-SANT domain to mediate interaction with the HDAC catalytic subunit. The SANT domain creates a binding site for inositol phosphates at the interface with the HDAC which controls activation of the enzyme[3,27].

In the MiDAC complex, the large ELM2-SANT protein MIDEAS (1045 aa), recruits HDAC1 or 2. The third component of the complex is the relatively small protein DNTTIP1 (329 aa) which contains both a dimerisation domain and DNA-binding domain[26]. Our co-immunoprecipitation studies clearly support an in vivo interaction between the three proteins in the complex. The proteins also appear to be expressed, co-localise and co-immunoprecipitate throughout the cell cycle. This is in apparent contrast to the findings of Bantscheff et al., who showed the complex was purified on HDAC-inhibitor coupled resin specifically when cells were blocked in mitosis using nocodazole[12]. This could be explained if the active site in the MiDAC complex is more accessible in cells during mitosis.

siRNA depletion of either MIDEAS or DNTTIP1 suggests that the MiDAC complex does indeed play a role in cell division. In both knockdowns, we observed a significant increase in chromosome misalignment during mitosis presumably due to loss of efficient chromosome congression mechanisms. HDAC complexes have been previously found to be required for normal mitosis. For example, we have previously observed that

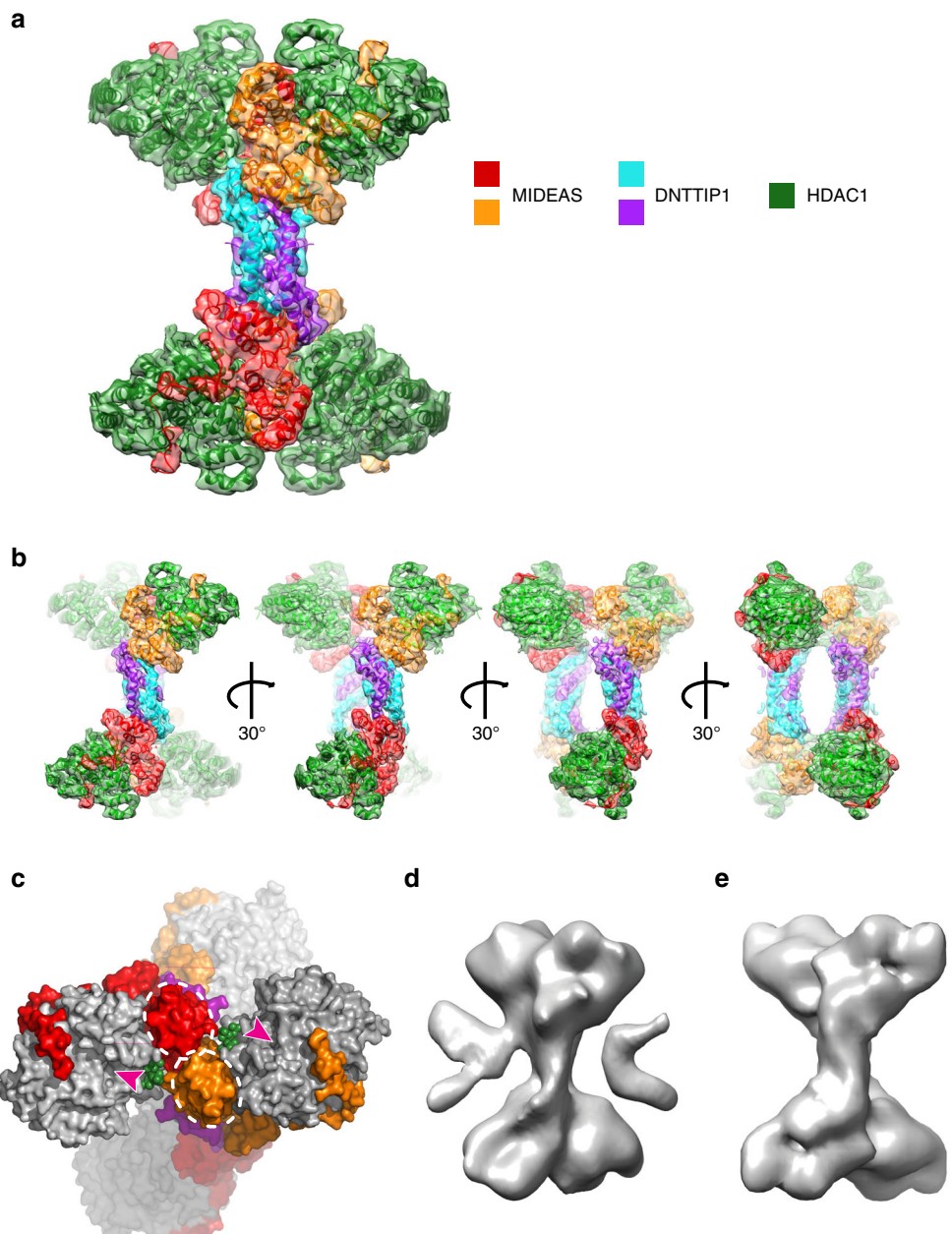

**Fig. 6 The structure of the tetrameric MiDAC complex. a** Post-processed cryo-EM map of the tetrameric complex from Relion3 with the protein chains shown as cartoons. The map is coloured according to the protein. Map contour level is 0.015—Chimera. **b** 30° rotations of the MiDAC tetramer with enhanced depth-cue. **c** End-on view of the MiDAC illustrating the interactions between the MIDEAS-SANT domains and InsP6 at the tetramerisation interface. The InsP6 is shown in green, the HDACs in grey surface and the positions of the HDAC1 active sites are indicated with magenta arrowheads. The MIDEAS-SANT domains are indicated by white dotted ovals. **d** Cryo-EM map of the complex with the DNA-binding domain from DNTTIP using C1 symmetry and no mask. **e** Cryo-EM map of the complex with the DNA-binding domain from DNTTIP using D2 symmetry and a mask.

knockouts of HDAC1 and/or HDAC2 in mouse ES cells results in lagging chromosomes in the anaphase, formation of micronuclei and monopolar spindles[31]. It has also been reported that knockout of HDAC2 in mouse embryos results in misaligned chromosomes and reduced kinetochore function via increased H4K16 acetylation during oocyte maturation[32]. Others have shown that the HDAC3 NCoR complex localises to the mitotic spindle and knockdown of HDAC3 resulted in a collapsed mitotic spindle and chromosome alignment defects[33]. It was proposed that this could be the result of increased H3K4 acetylation at centromeres preventing normal di-methylation and protein recruitment[34] or alternatively deacetylation of

nuclear distributed protein C (NuDC), a protein that regulates mitotic progression[35].

Interestingly, siRNA depletion of either MIDEAS or DNTTIP1 resulted in degradation of the other protein, suggesting that the three-way interaction is obligate. This interdependency is supported by multiple lines of evidence. First, the mouse deletions show that the embryos die at the same stage with exactly the same phenotype. Second, the Cancer Dependency Map shows a strong co-dependency for DNTTIP1 and MIDEAS (https://depmap.org/portal/). Finally, the cryo-EM structure of the MiDAC complex shows that interactions between MIDEAS and DNTTIP1 are

essential to establish the striking tetrameric X-shaped architecture.

Although DNTTIP1 is unique in the human genome, there are two genes that encode proteins with significant homology in the central ELM2-SANT domain of MIDEAS. These are TRERF1 which is expressed in most tissues, and ZNF541, which is expressed almost exclusively in testes. It is likely that ZNF541 and TRERF1 assemble alternative versions of the complex. Similarly, we have shown that both HDAC1 and HDAC2 are present in MiDAC complexes in U2OS cells and are likely to co-exist in the same complex. This fits with previous proteomic studies which show that MIDEAS interacts with both HDAC1 and HDAC2[12–14]. Hein et al. also showed that tagged MIDEAS pulls down the homologous protein TRERF1 so it is possible that MIDEAS and TRERF1 co-exist in the same complex[14]. ZNF541 is not observed in these proteomic experiments, which is to be expected since its expression is restricted to the testes.

The near-atomic resolution cryo-EM structure does not include the DNA-binding domain of DNTTIP1. It also lacks the N-terminal (716 aa) and C-terminal (158 aa) regions of MIDEAS that are predicted to be in large part disordered[36]. The dimerisation domain of DNTTIP1 is critical for the assembly of the complex since it interacts with the ELM2 domain of MIDEAS and dictates the relative positioning of the HDACs in the dimer. This dimerisation domain is unique to DNTTIP1 such that to date, no other proteins in HDAC complexes or elsewhere have been found to have a similar structure (Pfam name: DNTTIP1_dimer). Tetramerisation of the complex is mediated by the back surface of the MIDEAS-SANT domain. Interestingly, the tetrameric arrangement brings the inositol phosphate-binding sites on the two dimers in close proximity such that two conserved basic residues in the SANT domain of one MIDEAS are able to interact across the tetramerisation interface with the inositol phosphate in the adjacent dimer (Supplementary Fig. 10e). This would suggest that inositol phosphates may be particularly important for MiDAC activity.

Whilst the MiDAC complex is unique in having four HDAC subunits, several other class I HDAC complexes are also multimeric. The MTA proteins in the NuRD complex forms a dimeric architecture with two copies of HDAC1/2[3,37]. Analogously, SDS3 in the Sin3 complex mediates dimerisation assembling two HDAC1/2 enzymes[38]. The SMRT/NCoR complex is built round a tetrameric TBL1 scaffold that again recruits two copies of HDAC3[2,11]. We suggest that the relative orientation and spacing of the HDAC active sites is likely to be important for the function and specificity of these complexes such that the relative orientations of the multiple HDAC enzymes in the different complexes might target different conformations of chromatin. Certainly, the multivalent nature of the MiDAC complex is highly likely to be important for the efficient and processive deacetylation of chromatin domains so as to decommission gene promoters and enhancers. Given the overall size of the MiDAC complex and the outward-facing HDAC active sites at the complex extremities, it is likely that MiDAC is able to simultaneously target multiple nucleosomes. These distinct features of the MiDAC complex may be related to the fact that it appears to lack regions that direct recruitment to specific transcription factors and likely targets chromatin directly with the DNTTIP1 DNA-binding domain.

Our low-resolution structure of the larger MiDAC complex containing the DNA-binding domains of DNTTIP1 strongly suggests that they are flexibly linked to the core complex and are positioned around the waist of complex. We have previously shown that the MiDAC complex seems to prefer binding to linker DNA in chromatin[26]. It is attractive to imagine that these domains recruit the complex to open chromatin and that the flexible linker enables the core tetramer to deacetylate adjacent nucleosomes or nucleosomes in the local proximity. Such flexibly attached DNA-binding domains are a common theme in chromatin-targeted complexes e.g., the telomere-binding complex[30].

To understand the physiological function of MiDAC, we used CRISPR to create mice lacking either MIDEAS or DNTTIP1. Importantly, both $Mideas^{-/-}$ and $Dnttip1^{-/-}$ homozygotes die at the same developmental stage and with identical failure in heart development and haematopoiesis. These findings are important since they reveal that the MiDAC complex is essential for life and the loss cannot be compensated for by other complexes containing class I HDACs. The identical phenotype and gene expression changes of the two knockouts reinforces the finding that MIDEAS and DNTTIP1 are mutually interdependent. They also confirm that the chromosome misalignment phenotype of the knockdowns in cancer cell lines is due to loss of the MiDAC complex. Finally, the RNAseq data support the concept that MiDAC is a gene-regulatory complex, which has not been previously demonstrated.

The co-dependency of MIDEAS and DNTTIP1 fits well with the architecture of the complex revealed in the cryo-EM structure. However, it should be noted that the embryos survive to day e16.5 with many tissues appearing normal. This suggests that the increased mitotic chromosome misalignment phenotype observed in cell lines (and MEFs derived from the homozygous embryos) is not sufficient to perturb normal cell proliferation and differentiation. It is likely therefore that the failure of heart development and haematopoiesis is a more specific gene-regulatory defect. Other class I HDAC complexes have also been shown to be essential for embryo development including the NuRD, CoREST, SMRT/NCoR and Sin3 complexes. The cause of death and the stage at which the embryos die varies depending on the complex that is deleted, but are generally earlier than the MiDAC knockouts, suggesting that MiDAC may have a more specialised, although still essential function[39–43]. Knockouts of the $C. elegans$ proteins SAEG-1 and SAEG-2 (orthologues of MIDEAS / TRERF1 and DNTTIP1, respectively) are not lethal but do cause defects in body length and other behavioural abnormalities[44].

Transcriptomics in MEF cells derived from wild-type and both $Mideas^{-/-}$ and $Dnttip1^{-/-}$ mutant mice revealed that 468 genes are perturbed in both knockouts. Strikingly, the direction and magnitude of the change in RNAs from these overlapping genes are highly similar in the two knockouts. The majority of the perturbed genes are upregulated, consistent with the knockout of a transcriptional repression complex. Many of the perturbed genes/pathways are key for normal development, explaining the multiple developmental defects observed. However, the exact molecular mechanisms underlying these gene expression perturbations remain to be elucidated. Interestingly, Panther analysis identified a number of cell-cycle-associated genes including members of the septin family of GTP-binding proteins which have been implicated in chromosome alignment[45]. The trend towards an overall decrease in spindle organising proteins as seen from GSEA could provide an additional explanation for increased misalignment in the MIDEAS and DNTTIP1 knockout MEFs.

In conclusion, our structural and functional studies of the MiDAC complex reveal a striking molecular machine whose architecture appears to have evolved to facilitate processive chromatin deacetylation. The complex is clearly important for the regulation of many developmental genes and is essential for embryonic development.

## Methods
**Protein expression in HEK293F cells.** The proteins of the MiDAC complex, DNTTIP1, MIDEAS and HDAC1, were expressed using the pcDNA3 vector in HEK293F suspension mammalian cells (Invitrogen). For purification of the

complex, the MIDEAS construct was expressed with an N-terminal 10xHis-3xFLAG tag and a TEV protease cleavage site. Transient transfections were used to express the MiDAC complex, which involved mixing 0.1 mg of each plasmid (0.3 mg DNA total) with 30 ml of PBS (Sigma) and then adding 1.2 ml of 0.5 mg/ml PEI (Sigma). The suspension was vortexed briefly, incubated for 20 min at room temperature, then added to 300 ml of cells at a density of $1 \times 10^6$ cells/ml. The transfected cultures were harvested 48 h after transfection.

**Large-scale purification of the HDAC1/MIDEAS/DNTTIP1 complex.** HDAC1/MIDEAS/DNTTIP1 complexes were purified from 1.2 l of HEK293F cells. After sonication in a buffer containing 50 mM Tris/Cl pH 7.5, 100 mM potassium acetate, 10% (v/v) glycerol, 0.5% (v/v) Triton X-100 and Complete EDTA-free protease inhibitor (Roche) (buffer A), the insoluble material was removed by centrifugation. The complex was then bound to anti-FLAG resin (Sigma), washed twice with buffer A; three times with buffer B (50 mM Tris/Cl pH 7.5, 50 mM potassium acetate, 5% (v/v) glycerol, 0.5 mM TCEP); incubated with 0.5 mg RNaseA for 1 h at 4 °C and then washed five times with buffer B. TEV protease was then used to release the MiDAC complex from the resin overnight on a roller at 4 °C. The complex was gel filtrated on a Superdex-200 column (GE Healthcare) in 25 mM HEPES pH 7.5, 50 mM potassium acetate, 0.5 mM TCEP before making EM grids.

**Cryo-EM grid preparation.** The peak fraction (0.5 ml) from the Superdex-200 column was concentrated to 0.8 mg/ml in a 0.5 ml Amicon Ultra centrifugal filter with a 10 kDa MWt cutoff. After concentration, the complex was placed on ice. In total, 50 μM SAHA (an HDAC inhibitor) and 50 μM InsP6 (an HDAC regulator) were added, followed by an equal volume of 0.25% glutaraldehyde. After 20 min on ice, the cross-linking was stopped with the addition of Tris/Cl pH 7.5 to 50 mM. Then 3 μl sample was applied to a 1.2/1.3 UltrAuFoil grid. The grid was blotted for 3 s, before plunge freezing, using a ThermoFisher Vitrobot MKIV at 4 °C and 100% relative humidity, and storing in liquid nitrogen.

**Cryo-EM data collection.** Cryo-EM datasets were collected on a ThermoFisher Titan Krios G3 (EPU 1.9) operated at 300 kV with a Gatan Quantum Energy filter camera and a Volta Phase Plate with a defocus of −0.5 using a Falcon 3 camera with a nominal magnification of ×75,000 and a calibrated pixel size of 1.08 Å. A dose rate of 0.68 electrons per pixel per second in counting mode over 60 s and 75 fractions were used.

**Data processing.** Relion3.0 was used to process the data[46] (Supplementary Figs. 7 and 8). Particles were autopicked after creating references from manually picked particles. 2D class averaging was used to obtain an initial cleaned particle set. It was clear at this stage that the particles could be sub-categorised into dimer and tetramer. Further 2D classification was used to split each dataset into dimer and tetramer particles. After 3D classification, Refine3D, 3D classification without alignment, Refine3D, particle polishing and CTFrefine, a 4-Å map was obtained for the dimer and a 4.5-Å map for the tetramer.

**Map fitting and refinement.** The crystal structures of the HDAC1 and the dimerisation domain of DNTTIP1 could be unambiguously fitted into the EM maps for both the dimer and the tetramer using Chimera. An iTASSER model for the MIDEAS generated based on the crystal structure of the MTA1 in the HDAC1:MTA1 complex provided a starting point to rebuild the MIDEAS protein. The MIDEAS model was rebuilt in Coot[47] using a sharpened and blurred map generated using MRCtoMTZ in CCPEM[48]. In order to facilitate the rebuilding of the MIDEAS protein, a map masked with the HDAC1 and DNTTIP1 dimerisation domain was generated. After model rebuilding of the MIDEAS and fitting of the HDAC1 and DNTTIP1 crystal structures, the model was refined once in Phenix[49] using a rigid body refinement followed by one round of simulated annealing to minimise the clashes. Figures were prepared using either Chimera or MacPymol.

**Animals.** C57BL/6J mice were maintained in a specified pathogen-free (SPF) facility and used for mating between 6- and 26-weeks old. Mice were housed in individually ventilated cages on a 12 h day–night cycle in a temperature (21 ± 2 °C) and humidity-controlled (55 ± 10%) room. All mice had free access to water and were fed on the irradiated Teklad rodent diet (Envigo). All animals were kept in pairs and cages contained bedding made of corn cob with a nestlet (Datesand) and a cardboard tunnel for environmental enrichment. When breeding, one male was housed with two females until pregnancy was apparent, either by vaginal plug or an enlarged abdomen, when the female was separated. As a phenotype for heterozygous and homozygous knockouts was not known, in this study we sought to determine the viability of such mice using non-invasive genotyping from ear snips already taken for identification. Mice were culled by an approved technician by schedule 1 (dislocation of neck followed by cutting the femoral artery). Mouse embryos were culled by a trained technician by immersion in ice-cold PBS followed either by decapitation or submersion in fixative.

**Generation of CRISPR mice.** MIDEAS and DNTTIP1 knockout mice were generated using CRISPR/Cas9 system[50]. crRNA (IDT) (20 ng/μl) designed to target either exon 2 of *Mideas* (ENSMUSE00000408326: TCCCTACTATAACCACCC GGAGG) or *Dnttip1* (ENSMUSE00000171721: AACATCGGCAGGTGCAGCG AAGG), 20 ng/μl tracrRNA and 20 ng/μl of Cas9 protein (IDT) were injected into 1-cell C57BL/6J mouse zygotes under standard micro-injection conditions. The resulting pups were analysed for modified alleles by PCR and then Sanger sequencing. Mosaic founders were back-crossed to wild-type mice to segregate alleles, resulting in −10-bp (*Mideas*) and −11-bp (*Dnttip1*) deletion mutants.

**DNA extraction and genotyping.** When only genotyping was required, both mice and embryos were genotyped from DNA isolated from the ear and tail snips, respectively, using the HotSHOT method[51]. Briefly, alkaline lysis buffer (0.07% NaOH, 0.7 mM EDTA pH 8) was added to tissue samples and incubated at 95 °C for 30 min followed by 15 min at 4 °C. The lysis buffer was neutralised with an equal volume of 40 mM Tris-HCl pH 5. For histology, whole embryos were fixed in 10% formalin for 48 h. Before processing and embedding, a tail snip was taken, and DNA extracted using a previously described method[52]. Alkali digestion buffer (0.1 M NaOH, 1% SDS, pH 12) was added to tissue samples and incubated at 100 °C for 40 min. The sample was allowed to cool to room temperature before the addition of 25:24:1 phenol:chloroform:isoamyl alcohol. The sample was agitated for 5 min at room temperature and centrifuged at 10,000 $g$ for 5 min. The upper aqueous layer was transferred to a new tube with chloroform, agitated for 5 min at room temperature and centrifuged as above. The upper aqueous layer was transferred to a new tube along with 0.6 volumes isopropanol and 0.1 volume 3 M sodium acetate, pH 5. The solution was mixed briefly before centrifugation at 10,000 $g$ for 30 min at room temperature. The supernatant was decanted, and the pellet rinsed twice in 85% ethanol with centrifugation at 10,000 $g$ for 5 min between washes. Ethanol was removed by a brief incubation at 60 °C and the pellet resuspended in 50 μl TE buffer (10 mM Tris-HCl, pH 8, 0.1 mM EDTA). Isolated DNA was then used for genotyping by PCR using DreamTaq green PCR master mix (ThermoFisher). Wild-type and mutant-specific primers for MIDEAS-del1 mice, WT: 318-bp (F: 5′-CTATAACCACCCGGAGGCAC-3′, R: 5′-GAAGGCAGTTGATGCATGG-3′) or 182-bp mutant (F: 5′-ACCTCCCTACTATAACCACTGA-3′, R: 5′-AAGACCTG ACGGTTCACCTG-3′); DNTTIP1-del1 mice, WT: 220-bp (F: 5′-AGATCGGCG GCCCCTTCGCT-3′, R: 5′-GCGAGCTTTGGACATTGGTG-3′) or 351-bp mutated allele (F: 5′-GTCATCTGAGATCGGCGGCA-3′, R: 5′-AGCAATAACCCGAG CTTGCT-3′) were used. PCR amplification: 35 cycles of 95 °C for 30 s, 60 °C for 30 s and 72 °C for 1 min.

**Preparation of embryo sections for histology.** Mouse embryos were fixed in 10% formalin for 48 h before processing using a Leica ASP300 processor. Briefly, embryos were incubated for 1 h in 10% formalin followed by ×7 1-h incubations with 99% IMS, ×2 1.5-h incubations with xylene and ×1 1-h and ×2 1.5-h incubations in wax baths. Processed embryos were oriented in metal moulds and embedded in wax. A microtome cut 4-μm sections of embryos for further staining. Haematoxylin and eosin staining was automated using a Leica ST4040 Linear Stainer with a standard protocol. Briefly, sections were washed three times with xylene followed by a wash with 99% IMS and 95% IMS. After one wash with water, sections were stained with haematoxylin. After a further three washes with water, slides were stained with eosin. Sections were then washed in the opposite order to that described above. Slides were mounted using DPX and imaged using a Leica M165 or Nikon eclipse TI microscope.

**Generation of mouse embryonic fibroblasts (MEFs).** Timed mates were set up as above and females culled by schedule 1 at e13.5. Embryos were removed from the yolk sacs and washed three times in PBS. Embryo heads, liver and heart were eviscerated, and the remaining body minced in trypsin. Cells were left for 45 min at 37 °C 5% $CO_2$ before addition of complete DMEM. Cells were pipetted 15 times to form a single-cell suspension and transferred to a fresh 10-cm dish and cultured until confluent. MEFs were cultured for up to eight passages before reaching senescence.

**Cell culture.** Unless otherwise stated, U2OS osteosarcoma (ATCC, #HTB-96) and HeLa adenocarcinoma (ATCC, #CCL-2) cells were cultured in DMEM/10% foetal calf serum (FCS)/1% penicillin/streptomycin (P/S) (complete DMEM). For RNAi experiments, cells were seeded at $3 \times 10^5$ in Opti-MEM®/10% FCS. Embryonic stem (ES) cell lines were maintained on gelatinised plates in standard ES cell medium consisting of Knockout DMEM, 15% FCS, 1× glutamine/P/S, 100 μM β-mercaptoethanol and LIF (synthesised in house). Synchronisation was confirmed by flow cytometry. Cells were blocked by the addition of aphidicolin (1.6 μg/ml) (G1/S) or nocodazole (3.3 μg/ml) (G2/M) for 16 h and synchronised using the CDK inhibitor (CDKi; Ci) RO-3306 (10 μM) for 14 h followed by 2 h with nocodazole (30 ng/ml) before release. For protein expression, 293F cells were cultured in Freestyle media, as previously described[53]. All cells were cultured at 37 °C in a 5% $CO_2$ atmosphere.

**Generation of MIDEAS-FLAG in ES cells.** CRISPR/Cas9 (sgRNA: GCCCGGCA ATAGGATCCAGCAGG) was used to modify exon 13 of the mouse *Mideas* gene

in E14 embryonic stem (ES) cells. The stop codon was replaced with 3× copies of the FLAG epitope. Following co-transfection of the sgRNA/Cas9 plasmid (10 μg) and ssODN (5 μg), 500 cells were plated onto a 10-cm dish, cultured for 10 days, and the resulting colonies screened for the modified allele using PCR and then western blotting.

**Transient transfection of GFP-MIDEAS and mCherry-DNTTIP1.** U2OS (~8 × $10^5$) were seeded using complete DMEM into six-well plates and allowed to adhere overnight at 37 °C 5% CO$_2$. Plasmids and lipofectamine® were diluted in Opti-MEM® and incubated for 5 min at room temperature. The separate solutions were mixed and incubated for a further 20 min at room temperature before being added dropwise to the U2OS cells. Cells were incubated at 37 °C for 48 h in 5% CO$_2$ to allow for protein expression. Cells were then fixed for immunofluorescence or lysed directly with SDS sample buffer prior to western blotting.

Oligonucleotides used to generate GFP-MIDEAS and mCherry-DNTTIP1 constructs are provided in Supplementary Table 6.

**Generation of FLAG-DNTTIP1 stable cell line.** To generate a stable doxycycline (DOX) inducible siRNA-resistant FLAG-DNTTIP1 U2OS cell line, primers were designed to incorporate two silent mutations in the siRNA target sequences directed against DNTTIP1. These were introduced sequentially using PCR. A full-length DNTTIP1 PCR product with an N-terminal FLAG tag incorporating mutations in both sequences was cloned into a PiggyBac vector that contained a puromycin selection gene. The PiggyBac vector and a vector encoding transposase (Systems Bioscience, #PB210PA-1) were transiently transfected into U2OS cells using lipofectamine®. After 24 h, 80,000 cells were seeded into a 10-cm dish and grown in complete DMEM with 0.8 μg/ml puromycin. A minimum of 12 individual colonies were expanded and induced for 24 h with 1 μg/ml DOX. FLAG expression was tested by western blotting.

Oligonucleotide's used to generate DNTTIP1 siRNA-resistant DOX inducible U2OS cell line are provided in Supplementary Table 6.

**Knockdown of MIDEAS and DNTTIP1 using siRNA.** Two siRNAs directed against either MIDEAS or DNTTIP1 (see below) were transfected into cells using Oligofectamine™ reagent. Briefly, in separate tubes, individual siRNA or Oligofectamine™ was diluted in Opti-MEM® and incubated at room temperature for 5 min. The siRNA/Opti-MEM® mixture was added to the Oligofectamine™ and incubated at room temperature for a further 20 min. Cells that had been seeded the day before in Opti-MEM® 10% FCS were transferred to just Opti-MEM®, and the transfection mixture was added dropwise. After incubation for 6 h, Opti-MEM® with 30% FCS was added and the cells incubated for a further 72 h to achieve knockdown.

siRNA sequences and sources used in this study:

- MID_si1–5′-UCUGAAGAGAAGCGGAAAA-3′ (Dharmacon™, #J-031938-19)
- MID_si2–5′-AAACAGAGACAUUCAGUAAtt-3′ (Ambion®, #s40760)
- DNT_si1–5′-CCCACACCUCUUUAAGUAUtt-3′ (Ambion®, #s41923)
- DNT_si2–5′-CCUUGGAACAUAAUGAUAAtt-3′ (Ambion®, #s41294).

**Real-time qPCR.** For relative mRNA expression by qPCR, RNA was purified from cells using the quick RNA mini prep kit (Zymo Research). Briefly, cells were lysed in lysis buffer, cleared by centrifugation and gDNA removed using a spin-away™ filter. An equal volume of 100% ethanol was added to the flow through and transferred to a Zymo-Spin™ IIICG column. The RNA was bound to the column, washed once with RNA wash buffer and the membrane incubated with DNase 1 for 15 min. The column was washed once with RNA prep buffer followed by two washes with RNA wash buffer before elution with 100 μl DNase/RNase-free water. The concentration and quality of the RNA were analysed using a nanodrop. cDNA was synthesised from 1 μg of the total RNA using 5× qScript reaction mixture and qScript reverse transcriptase (Quanta Bioscience). cDNA was synthesised using the following programme: 22 °C, 5 min; 42 °C, 30 min; 85 °C 5 min; 4 °C. cDNA was diluted 1 in 5 and stored at −20 °C. Real-time qPCR was carried out using the SensiFAST™ kit (Bioline) with 400 nM forward and reverse primer and 2 μl cDNA. In this study, primers for human MIDEAS (Qiagen; QT00076160), DNTTIP1 (Qiagen; QT00069664), HDAC1 (Qiagen; QT00015239) and the housekeeping gene β2-microglobulin (Qiagen; QT00088935) were used and PCR product amplified with 35 cycles of 95 °C for 10 s, 60 °C for 30 s and 72 °C for 1 min using a CFX Connect™ real-time system (Bio-Rad).

**Cell-cycle analysis.** To determine cell-cycle distribution, cells were washed once with PBS before fixation with −20 °C 70% (v/v) ethanol at 4 °C for 30 min. Cells were washed two times with PBS and incubated with 50 μg propidium iodide (PI) and RNaseA (10 μg/ml, thermoFisher) for 16 h at 4 °C. Samples were analysed using a BD Accuri™ C6 flow cytometer (BD Biosciences) and FCSalyzer software v0.9.15-alpha. Gating strategies used for each cell type are shown in Supplementary Fig. 13.

**Immunofluorescence microscopy.** Cells were grown on acid-treated coverslips and fixed with −20 °C methanol or 4% paraformaldehyde (PFA) followed by permeabilisation with 0.2% Triton X-100/PBS. Following fixation/permeabilisation, cells were washed three times in PBS and blocked with 2% BSA/PBS for 30 min. All subsequent antibody incubations were carried out in PBS with 3% BSA. Primary antibodies used in this study were rabbit anti-MIDEAS (1:50; Atlas #HPA003111), rabbit anti-DNTTIP1 (1:100; Abcam #ab174663), rabbit anti-HDAC1 (1:200; Abcam #ab109411), anti-α-tubulin (rabbit: 1:1000, ThermoFisher #PA5-19489; mouse: 1:2000, Sigma #T9026) and mouse anti-cenpA (1:1000; Abcam #ab13939). Cells were dual stained for 1.5 h and then washed three time with PBS. Primary antibodies were detected with goat anti-rabbit 488 and goat anti-mouse 594 antibodies (1:200; Invitrogen #A32731 #A32742, respectively). DNA was stained with Hoechst 33258 (0.2 μg/ml). After two final washes in PBS followed by two washes in dH$_2$O, coverslips were mounted in 80% glycerol, 3% n-propylgalate mounting medium. Cells were imaged using a Leica SP5 laser-scanning confocal microscope with the 63x oil immersion lens. Chromosome misalignment was defined when microtubules were perpendicular to condensed chromosomes with co-localisation of DNA and cenpA outside the microtubule region.

**Isolation of cytosolic and nuclear proteins.** Cytosolic, soluble nuclear and insoluble nuclear proteins were isolated using a commercial kit (Abcam #ab219177) following the manufacturer's instructions. Briefly, cells were washed once in ice-cold PBS and lysed with cytosolic extraction buffer. After incubation on ice, nuclei were pelleted by centrifugation at 1000 g for 3 min at 4 °C, and cytosolic proteins transferred to a clean microfuge tube. Nuclei were lysed using soluble nuclear lysis buffer and the suspension incubated on ice for 15 min, vortexing every 5 min. Insoluble material was pelleted by centrifugation at 5000 g for 3 min at 4 °C, and the soluble nuclear fraction transferred to a clean microfuge tube. Insoluble nuclear lysis buffer was added to the DNA:protein pellet and disrupted by intermittent sonication (10 s on, 10 s off). Isolated protein fractions were stored at −80 °C. Each buffer was supplemented with 200× protease inhibitor cocktail and DTT supplied with the kit.

**Immunoprecipitation.** Protein A or G Dynabeads® (1.2 mg; Life Technologies) were washed with lysis buffer and incubated with primary antibodies for 30 min at 4 °C. In this study, primary antibodies used were rabbit anti-MIDEAS (Atlas), rabbit anti-DNTTIP1 (Abcam), a rabbit polyclonal IgG control (Abcam #ab37415), mouse anti-FLAG (Sigma #F3165) and a mouse monoclonal IgG control (Santa Cruz #sc-2025) (all 1 μg). In some cases, IgG or antibody was covalently coupled to Dynabeads® using BS3 (ThermoFisher #A39266). Antibody-coupled Dynabeads® were washed once with conjugation buffer (20 mM sodium phosphate, 0.15 M NaCl, pH 8) and suspended in conjugation buffer with 5 mM BS3. The antibody/Dynabeads® suspension was rotated for 30 min at room temperature and cross-linking quenched by addition of 1/20 volume 1 M Tris/HCl, pH 7.5 and rotation for 15 min at room temperature. The cross-linked Dynabeads®/antibodies were washed three times in lysis buffer before proceeding with the IP. Protein lysates were added to the antibody-coupled Dynabeads® and incubated at 4 °C for 16 h with gentle mixing. Immunoprecipitates were washed three times in lysis buffer and re-suspended in an appropriate amount of buffer for downstream applications.

**SDS-PAGE and western blotting.** Separation and detection of proteins by SDS-PAGE and western blotting was carried out using NuPAGE 4-12% Bis-Tris gels (Invitrogen) and a semi-dry transfer system using nitrocellulose membranes. Membranes were blocked for 1 h at room temperature using 5% milk TBS/T and incubated in primary antibody diluted in blocking buffer overnight at 4 °C. For western blotting, antibodies used were rabbit anti-MIDEAS (1:300, Atlas), rabbit anti-DNTTIP1 (1:500, Abcam), anti-HDAC1 (rabbit: 1:500, Abcam; mouse: 1:400, Santa Cruz), mouse anti-HDAC2 (1:1000, Sigma, #05-814), mouse anti-FLAG (1:2000, Sigma), goat anti-LaminB (1:1000, Santa Cruz) and rabbit anti-α-tubulin (1:1000, ThermoFisher). Goat anti-mouse HRP (1:2000, #12-349), goat anti-rabbit HRP (1:2000, #12-348), donkey anti-goat HRP (1:5000, #AP180P) (all from Sigma), goat anti-rabbit 800CW (1:10000, #925-32211) and goat anti-mouse 680RD (1:10,000, #925-68070) (both from Li-COR, IRDye®) secondary antibodies were incubated with membranes for 1 h at room temperature and detected using either enhanced chemiluminescence (ThermoFisher) or an Odyssey CLx digital imaging system (LI-COR).

**Histone deacetylase activity assay.** HDAC activity was measured using Boc-Lys (Ac)-AMC substrate (BaChem #4033972). Immunoprecipitated complexes were left bound to the Dynabeads® without cross-linking of antibodies. In some cases, immunoprecipitates were pre-incubated in the presence or absence of either InsP6 (100 μM) or SAHA (5 μM) for 30 min at room temperature with gentle agitation. Immunoprecipitates and Boc-Lys(Ac)-AMC substrate (500 μM final concentration) were used in a final volume of 50 μl in lysis buffer. After incubation at 37 °C for 30 min and 200 rpm, the assay was developed with the addition of 50 μl of developer solution (50 mM Tris/HCl pH 7.5, 100 mM NaCl, 2 mM Trichostatin A, 10 mg/ml Trypsin). Fluorescence was measured at 335/460 using a Victor X5 plate reader (Perkin Elmer). Technical duplicates were performed a minimum of three

times, and the data were analysed using GraphPad Prism (version 7.0, GraphPad Software, Inc.).

**Generation and analysis of RNAseq data**. RNAseq data were collected using total RNA from wild-type and knockout MIDEAS and DNTTIP1 cell lines. Briefly, the total RNA was extracted from MEF cells using the quick RNA mini prep kit (described above), quality checked using a bioanalyzer and shipped at 2 μg on dry ice to Novagene (China). Samples were enriched for mRNA before double-strand cDNA synthesis. Adaptor sequences, containing an overhang, were ligated to the cDNA. After fragmentation, the library was quality tested before Illumina sequencing. Each sample was read to a depth of $20 \times 10^6$ reads. The data files were processed in house. Adaptor sequences were trimmed using the Trim Galore! wrapper for Cutadapt, with the default settings. Sequences shorter than 35 bp were removed. The trimmed reads were then aligned to the most recent *Mus musculus* genome (MM10), using the STAR aligner with the default settings. Transcripts were deduplicated, with Picard, sorted and indexed (SAMtools) before analysis using DESEQ2 software in R to generate principle component plots for the data. Comparisons of gene expression differences were carried out between wild-type samples and the MIDEAS and DNTTIP1 knockout samples. Differences were deemed significant with a *P*-value of <0.1.

**Study approval**. All animal experiments conformed to the British Home Office Regulations (Animals (Scientific Procedures) Act 1986). The project was approved by the Animal Welfare and Ethical Review Board (AWERB) at the University of Leicester under Project licence P16D64BDE (Dr Amanda Pickard) and PEBDF7FCB (Prof Shaun Cowley) following the principles of the 3Rs and ARRIVE guidelines.

**Statistical analysis**. All data graphs are presented as mean ± SEM, and accompanying statistics were generated using GraphPad Prism software. Statistical tests and replicate number for each individual experiment is stated in the accompanying figure legend.

**Materials availability**. Unique reagents generated in this study are available from the lead contact with a completed Materials Transfer Agreement.

**Reporting summary**. Further information on research design is available in the Nature Research Reporting Summary linked to this article.

## Data availability
The EM maps for the dimer and tetramer are available from EMDB under the accession codes EMD-11041 and EMD-11042. The coordinates for the dimer and tetramer models are available from the PDB under the accession codes 6Z2J and 6Z2K. The RNAseq data that support the findings of this study have been deposited in GEO with the primary accession code GSE144748. The source data underlying Figs. 1–3 and Supplementary Figs. 1, 3, 4, 5, 6, 9 and 12 are provided as a Source Data file. All other data are available from the corresponding authors on reasonable request.

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

## Acknowledgements

We are grateful to the PROTEX facility (University of Leicester) for preparation of expression clones; to the eBIC facility at Diamond for initial cryo-EM datasets (EM18094 & EM14273); to Richard Collins (University of Manchester) for help with early EM grid preparation and visualisation; to Laura O'Regan for training and support for confocal fluorescence imaging; to Aidan Michaels for cell biology in HeLa cells; to the Preclinical Imaging Facility (University of Leicester) for help in creating and maintaining mouse lines; to the University of Leicester Histology Service and to Martin Dyer, and particularly Peter Tontonoz, for advice on interpreting the histology. Initial electron microscopy studies were supported by Ian Hands-Portman and Warwick Life Sciences Imaging Suite (now Advanced Bioimaging Research Technology Platform), University of Warwick, using equipment funded by the Wellcome Trust (055663/Z/98/Z). We acknowledge the Midlands Regional Cryo-EM Facility at the Leicester Institute of Structural and Chemical Biology (LISCB), major funding from MRC (MC_PC_17136). This work was supported by a Wellcome Trust Senior Investigator Award to J.W.R.S. (WT100237/Z/12/Z); a Royal Society Wolfson Merit Award to J.W.R.S.; a BBSRC Project Grant to S.M.C. & J.W.R.S. (BB/N002954/1); an MRC Capital Award for the Cryo-EM Facility to J.W.R.S. (MC_PC_17136). S.M.C. was supported by a senior non-clinical fellowship from the MRC (MR/J009202/1).

## Author contributions

Conceptualisation: J.W.R.S., L.F., R.E.T. and S.M.C.; biochemistry: L.F., A.S. and O.V.M.; Cryo-EM sample preparation, data collection and processing: L.F., A.S., K.L.M., T.J.R., C.G.S., O.V.M., C.J.S. and A.M.R.; Cryo-EM model building: L.F., A.S., C.J.M., T.J.R., C.G.S. and J.W.R.S.; cell Biology: R.E.T., E.K., A.M.F. and S.M.C.; mouse knockouts: E.K., R.E.T., S.M.C. and J.W.R.S.; RNAseq analysis: R.E.T. and A.C., supervision: J.W.R.S., S.M.C., A.M.F. and C.J.S. Writing—original draft: L.F., R.E.T. and J.W.R.S. Writing—review and editing: all authors.

## Competing interests

The authors declare no competing interests.
