## [Peer Review File · Nature Communications]

Reviewers' comments:

Reviewer #1 (Remarks to the Author):

This manuscript reports a comprehensive functional and structural investigation of the MiDAC complex, the less studied of the class I HDAC complexes. The data presented convey important biological information, which should be of interest to a large audience.

However before publication the authors should pay attention to the following points.

1-Figure 1B shows that the DNNTIP1 immunoprecipitation brings down much less HDAC1 than MIDEAS immunoprecipitation. However, Figures 1C and D show that these two immunoprecipitations present comparable HDAC activities.

This discrepancy needs to be explained. Is the immunoprecipitation shown in Figure 1B representative of all IPs (including those shown in C and D)?

2-Figure 1F and 1G, it would have been more informative if the authors had shown the total distribution of cells along the cell cycle (i. e., FACS profiles) rather than time points (Fig. 1F) or indicating the % of cells only in one phase (Fig. 1G). Additionally, in Fig. 1G, the immunoprecipitated Flag-MIDEAS should be shown. This control is important since the total amount of DNNTIP1 in different phases of the cell cycle shown in Fig. 1F appears different from the co-immunoprecipitated DNNTIP1 shown in Fig. 1G. Indeed Co-immunoprecipitated DNNTIP1 appears more abundant in S and G1 than in G2/M (Fig. 1G), compared to the asynchronous cells or cells in G2/M.

3-Figure 1H, based on the data shown, the authors propose that the HDAC activity of the complex is higher in G1/S than the activity recovered from both asynchronous cells or in cells in mitosis.

However, in order to support this conclusion, they should have visualized the immunoprecipitated HDACs. Indeed, considering the possible variability in HDAC IP (Fig. 1B), the results could simply reflect the variable amounts of immunoprecipitated HDACs in the different cell types.

4-From Supp. Figure 2D we can see that the knock-down of DNNTIP1 and MIDEAS variably affects the total HDAC1 protein levels. In fact, these variations at the protein levels are somehow reminiscent of variations at the transcript levels shown in Supp. Figure 1C.

Would it be possible that HDAC1 gene expression is positively regulated by the complex?

Could the authors complete these data by using their transcriptomic data (Fig. 4)?

5-Legend Fig 2 E-F, Please define what the authors mean by “proteins were isolated from nuclear extracts”. Also the first line an “i” is missing in “wth”.

Page 8 top, “A” is missing. Should be Figure 3 A and B.

6-Figure 3 D the arrows are barely visible. Please change their size and colour.

7-The data shown in Supp. Figure 4 are somehow inconsistently presented for MIDEAS-del and DNTTPA1-del cells. Please present the same information for cells of both genotypes (genotyping, HDAC activity, cell growth, cell cycle distribution and westerns).

8-Where is the principal component analysis (PCA) described at the bottom of page 8 shown?

Reviewer #2 (Remarks to the Author):

The deacetylases HDAC1 and HDAC2 are subunits of multiprotein complexes. The association with components of these complexes is required for their activation, regulation and recruitment. The MiDAC complex was identified in a chemoproteomic approach as mitosis-specific HDAC1/HDAC2 complex but near to nothing is known about the function of this complex. In the first part of the paper the authors investigate the MiDAC proteins DNTTIP1 and MIDEAS. DNTTIP1, MIDEAS and HDAC1 are localized in the nucleus but are excluded from mitotic chromosomes. Knock-down of the proteins in human tumor cells caused the misalignment of mitotic chromosomes suggesting a role of the MiDAC complex in cell cycle regulation and maintenance of genomic integrity. Interestingly, the expression of both proteins is mutually co-dependent. DNTTIP1 and MIDEAS interact with each other throughout the cell cycle and are continuously associated with HDAC activity. In the second part of the manuscript the authors performed a CRISPR-Cas9-mediated knockout of DNTTIP1 and MIDEAS in mice. DNTTIP1 and MIDEAS KO mice die during late embryogenesis and have very similar phenotypes namely heart malformation and hematopoietic failure. Similar to the knock-down tumor cells DNTTIP1 and MIDEAS KO MEFs showed misalignment of mitotic chromosomes. However, proliferation and cell cycle progression was normal. Gene expression profiling revealed a similar

deregulation of genes in both KO cell lines suggesting a potential role of the MiDAC complex in transcription control. Finally, the authors determined the cryo-EM structure of the MiDAC complex, a tetrameric structure with the four HDAC1 enzymes positioned at the periphery with outward-facing active sites.

Criticisms:

In this manuscript the authors report interesting findings concerning the structure of the MiDAC complex and the phenotype of cells or mice lacking this complex. However, they do not provide insight into the mechanistic function of the MiDAC complex in mammalian cells to connect structural data to the mitotic phenotype or the developmental phenotype. For instance, histone hyperacetylation has been shown to disrupt centromeric heterochromatin and to affect kinetochore assembly and sister chromatid separation. Therefore, it would be important to investigate whether loss of MiDAC results in changes in histone acetylation and repressive histone methylation marks during mitosis in general and specifically at certain chromosomal regions such as centromeres. Given that the MiDAC complex could target multiple nucleosomes ChIP-seq experiments should be performed to examine whether MiDAC associates with specific chromosomal regions in the context of the mitotic phenotype or specific genes in the context of the deregulated genes.

In the current form the analysis of the KO mice adds little to the manuscript. The phenotype of MEFs is similar to the one of knock-down U2OS cells and the impact on mitosis does not affect early embryogenesis. The effects on heart development and the hematopoietic system are unclear and might be investigated in another study. In this context, loss of the MiDAC complex could be partially compensated in the early phase of development by a related HDAC complex, which might be absent in specific differentiated cell types.

In the first part the authors provide convincing data showing that the MiDAC complex is functional as deacetylase complex not only during mitosis but throughout the cell cycle. As discussed by the authors this is in contrast to previous findings and would suggest that the MiDAC complex has a more general function. Therefore it would be important to show that HDAC1 is associated with DNNTIP1 and MIDEAS during interphase.

Reviewer #3 (Remarks to the Author):

MiDAC is one of the class I histone deacetylases. The global structure and detailed mechanisms remain unclear. As other class I histone deacetylases, MiDAC is a multi-protein complex, composed of an HDAC catalytic subunit and two scaffold subunits, MIDEAS and DNNTIP1. In this manuscript, Turnbull, et al. designed a series of functional and structural studies to characterize the MiDAC complex and explore its functions. They determined the cryo-EM structures of MiDAC at resolutions of 4-5 angstrom, revealing a unique architecture different from other class I histone deacetylases.

Overall this work was well executed with a large body of data from a combination of different tools, including cell biology, animal model and structural biology. Before formal acceptance for publication, several issues should be addressed in the revision.

1. As stated in the last sentence in Abstract, “Four copies of HDAC1 are positioned at the periphery with outward-facing active sites so as to target multiple nucleosomes suggesting...”. “target multiple nucleosomes” is a hypothesis based on cryo-EM structures and not supported by any functional data in previous or this study. Therefore, the wording should also be more accurate.

2. In Figure 2A, the definition of y-axis, “Relative Density”, should be explained in legend. The reference value for normalization in the histogram was not clearly defined. Were the column heights normalized based on the Control lane? If so, why the height of Control lane in the top panel was more than 2? In addition, in the bottom panel, the protein concentrations in MID_si1/2 were significantly less than Control lane, but the column heights are similar.

3. The description in the second paragraph of Page 6. In Supplementary Figure 2A-B, the knock-down of MIDEAS or DNNTIP1 only suppressed the mRNA expression of their own. But in Figure 2A, the protein levels of both MIDEAS and DNNTIP1 were affected in either MIDEAS or DNNTIP1 depleting cell lines. It seems that the “mRNA expression” and “protein levels” were detected in the same manner using western blotting and the difference above accounted for the difference sampling: whole cell extract v.s. soluble nuclear fraction. This should be clarified in the revision.

4. In Figure 3C-D, the background and contrast in these images seemed to have been manually adjusted. If the adjustment is necessary, details should be provided in the legend.

5. What does the “c. 7 Å” in the first paragraph of Page 10 and other parts of the article mean? Please describe it upon the first usage of this term.

6. The structural data show a unique architecture for the MiDAC complex. However, many conclusions in the text were presented without supporting figures provided (listed below). The authors should provide a number of necessary supporting figures for better presentation of their structural data:

(1) The last sentence in Page 10, “Density for the InsP6, but not the SAHA, was clearly visible in the map.”;

(2) The second paragraph in Page 11, “The shorter helix in MIDEAS mediates a tight interaction with a non-polar groove between the end of the long helices of the DNNTIP1 dimer”;

“The SANT domains of MIDEAS and MTA1 are very similar in structure to each other and to the SANT domain of the SMRT protein that binds to HDAC32”;

“One of the regions of MIDEAS most similar to other ELM2-SANT domain corepressor proteins is the ELM2 specific motif which binds in an extended conformation in a conserved groove on HDAC1”.

(3) The first paragraph in Page 12, “The arrangement is such that the DNNTIP1 dimerisation domains are symmetrically juxtaposed, but do not directly interact.”;

“The two S-shapes are in opposite orientations such that the four HDAC subunits are positioned at the periphery, resulting in a striking three-dimensional X-shaped complex (Supplementary Figure 7H)”. “opposite orientations” should be clearly displayed.

(4) The second paragraph in Page 12, “The tetramerisation interface between the two dimers is mediated by interactions between the SANT domains”;

“Residues Lys849, Tyr874 and Gln877 contribute to the interface between the two neighbouring SANT domains”;

“These residues are not conserved in MTA1”;

“Interestingly, Lys839 and Lys843 appear to be in a position to interact with the inositol phosphate that is bound to the neighbouring dimer, although there is no apparent density for the side chains”;

“This may explain why it was only possible to obtain a high-resolution map of the tetramer in the presence of inositol phosphate (7 Å vs 4.5 Å)”. The image processing for “the tetramer in without inositol phosphate” should be described in Method session. Related figure should also be provided.

“A further obvious effect of the InsP6 was to significantly improve the local resolution of the core of the HDAC1, consistent with the proposed mechanism of activation”.

(5) The third paragraph in Page 12, “In addition to the interactions between the SANT domains it appears that the residues Nterminal to the DNTTIP1 dimerisation domain are crossing over to interact with the neighbouring tetramer”.

(6) The last paragraph in Page 12, “We also analysed cryo-EM grids of complexes containing full-length DNTTIP1 including the DNA-binding domain (Supplementary Figure 7A-D)”. The cited figures were incorrect.

(7) The first paragraph in Page 16, “Interestingly, the tetrameric arrangement brings the inositol phosphate binding sites on the two dimers in close proximity such that two conserved basic residues in the SANT domain of one MIDEAS are able to interact across the tetramerisation interface with the inositol phosphate in the adjacent dimer”.

7. In Supplementary Figure 5, cryo-EM maps of the intermediate processing steps, especially for the results of 3D classification, should be displayed here.

8. Judged by Supplementary Figure 6D, there seems to be a preferred orientation problem for the tetramer complex. Please provide a figure panel displaying the angular distribution of the final refinement.

9. The Data collection and Image processing for the data from grafix cross-linking, and gentle cross-linking samples without DNTTIP1 truncation could be included in Method section.

10. As stated in the last paragraph of Page 11, “The most ordered regions are the DNTTIP1 dimerisation domain and the core of HDAC1 (Figure 7A & B and Supplementary Figure 6)”. That means the regions of MIDEAS are less ordered? Since MIDEAS is located between DNTTIP1 and HDAC1, not the most peripheral, does the lower resolution correlate with its functional dynamics for

the complex? Have the authors tried to perform local 3D classification focused on the MIDEAS region to separate different conformations?

Reviewer #1 (Remarks to the Author):

This manuscript reports a comprehensive functional and structural investigation of the MiDAC complex, the less studied of the class I HDAC complexes. The data presented convey important biological information, which should be of interest to a large audience.

However before publication the authors should pay attention to the following points.

1-Figure 1B shows that the DNTTIP1 immunoprecipitation brings down much less HDAC1 than MIDEAS immunoprecipitation. However, Figures 1C and D show that these two immunoprecipitations present comparable HDAC activities.

This discrepancy needs to be explained. Is the immunoprecipitation shown in Figure 1B representative of all IPs (including those shown in C and D)?

The reviewer is correct that we have always found that the detection of HDAC1 in the IPs to be variable. We have now investigated this further have found that this is due to the HDAC1 antibody we have been using. We have now performed further IPs using a different HDAC1 antibody and also an antibody to HDAC2. We have performed these experiments in asynchronous cells and cells in G1/S and M phases (Figure 1G).

2-Figure 1F and 1G, it would have been more informative if the authors had shown the total distribution of cells along the cell cycle (i. e., FACS profiles) rather than time points (Fig. 1F) or indicating the % of cells only in one phase (Fig. 1G). Additionally, in Fig. 1G, the immunoprecipitated Flag-MIDEAS should be shown. This control is important since the total amount of DNTTIP1 in different phases of the cell cycle shown in Fig. 1F appears different from the co-immunoprecipitated DNTTIP1 shown in Fig. 1G. Indeed Co-immunoprecipitated DNTTIP1 appears more abundant in S and G1 than in G2/M (Fig. 1G), compared to the asynchronous cells or cells in G2/M.

These are good points and we have now included the FACS profiles for figures 1E-H in Supplementary figure 2.

With regards to the FLAG IP control, we have now included this in figure 1F to show the amount of protein pulled down. From the FLAG western we can see more FLAG pulled down in the S and G1 phases and so this likely accounts for the higher DNTTIP1 present in these pull downs.

3-Figure 1H, based on the data shown, the authors propose that the HDAC activity of the complex is higher in G1/S than the activity recovered from both asynchronous cells or in cells in mitosis.

However, in order to support this conclusion, they should have visualized the immunoprecipitated HDACs. Indeed, considering the possible variability in HDAC IP (Fig. 1B), the results could simply reflect the variable amounts of immunoprecipitated HDACs in the different cell types.

As mentioned in response to point 1 above, we have used a new antibody to detect HDAC1 and also used an additional antibody to detect HDAC2 (which is interchangeable with HDAC1). We now see consistent levels of HDAC1/2 being pulled down at these cell

cycle stages. Nevertheless, in the light of the results in figure 1G, we agree that we cannot firmly conclude that the activity of the complex is higher in G1/S. We have weakened this in the text.

4-From Supp. Figure 2D we can see that the knock-down of DNTTIP1 and MIDEAS variably affects the total HDAC1 protein levels. In fact, these variations at the protein levels are somehow reminiscent of variations at the transcript levels shown in Supp. Figure 1C.

Would it be possible that HDAC1 gene expression is positively regulated by the complex?

Could the authors complete these data by using their transcriptomic data (Fig. 4)?

This is a good suggestion. However, we have now re-visited the transcriptomic data from the RNAseq in MEFs derived from the KO mice. Knockout of either MIDEAS or DNTTIP1 has very little effect on HDAC1 mRNA levels (p-values 0.7; fold change -0.11 and +0.11 respectively).

5-Legend Fig 2 E-F, Please define what the authors mean by “proteins were isolated from nuclear extracts”. Also the first line an “i” is missing in “wth”.

Page 8 top, “A” is missing. Should be Figure 3 A and B.

We have also now referred to the Methods where we say that “proteins were isolated from nuclear extracts”.

Many thanks for pointing out these typos which we have corrected.

6-Figure 3 D the arrows are barely visible. Please change their size and colour.

Agreed. We have amended the arrows so they are more visible on the images

7-The data shown in Supp. Figure 4 are somehow inconsistently presented for MIDEAS-del and DNTTIP1-del cells. Please present the same information for cells of both genotypes (genotyping, HDAC activity, cell growth, cell cycle distribution and westerns).

Agreed. We have rearranged the figure so as to present equivalent data for the MIDEAS-del1 and DNTTIP1-del1 MEF lines side by side. Note that for panel D we are unable to do the IP using a MIDEAS antibody (equivalent to panel C) since the human MIDEAS antibody does not recognise the mouse protein and there is no mouse antibody available. Similarly, for the same reason, we cannot show the western blot for MIDEAS in panel C.

8-Where is the principal component analysis (PCA) described at the bottom of page 8 shown?

We had originally included the PCA plot in a draft manuscript but omitted it on discussion with an experienced colleague who suggested that it was not particularly useful since we would not expect to see tight clustering given that the MEF cell lines were prepared from embryos with different parents. Thus in addition to the biological and technical variation likely to occur when carrying out replicates, there will be

inherent variation between the developing embryos themselves. We have now removed reference to the PCA plot.

For the reviewer's information we show the plot below.

Reviewer #2 (Remarks to the Author):

The deacetylases HDAC1 and HDAC2 are subunits of multiprotein complexes. The association with components of these complexes is required for their activation, regulation and recruitment. The MiDAC complex was identified in a chemoproteomic approach as mitosis-specific HDAC1/HDAC2 complex but near to nothing is known about the function of this complex. In the first part of the paper the authors investigate the MiDAC proteins DNTTIP1 and MIDEAS. DNTTIP1, MIDEAS and HDAC1 are localized in the nucleus but are excluded from mitotic chromosomes. Knock-down of the proteins in human tumor cells caused the misalignment of mitotic chromosomes suggesting a role of the MiDAC complex in cell cycle regulation and maintenance of genomic integrity. Interestingly, the expression of both proteins is mutually co-dependent. DNTTIP1 and MIDEAS interact with each other throughout the cell cycle and are continuously associated with HDAC activity. In the second part of the manuscript the authors performed a CRISPR-Cas9-mediated knockout of DNTTIP1 and MIDEAS in mice. DNTTIP1 and MIDEAS KO mice die during late embryogenesis and have very similar phenotypes namely heart malformation and hematopoietic failure. Similar to the knock-down tumor cells DNTTIP1 and MIDEAS KO MEFs showed misalignment of mitotic chromosomes. However, proliferation and cell cycle progression was normal. Gene expression profiling revealed a similar deregulation of genes in both KO cell lines suggesting a potential role of the MiDAC complex in transcription control. Finally, the authors determined the cryo-EM structure of the MiDAC complex, a tetrameric structure with the four HDAC1

enzymes positioned at the periphery with outward-facing active sites.

Criticisms:

In this manuscript the authors report interesting findings concerning the structure of the MiDAC complex and the phenotype of cells or mice lacking this complex. However, they do not provide insight into the mechanistic function of the MiDAC complex in mammalian cells to connect structural data to the mitotic phenotype or the developmental phenotype. For instance, histone hyperacetylation has been shown to disrupt centromeric heterochromatin and to affect kinetochore assembly and sister chromatid separation. Therefore, it would be important to investigate whether loss of MiDAC results in changes in histone acetylation and repressive histone methylation marks during mitosis in general and specifically at certain chromosomal regions such as centromeres. Given that the MiDAC complex could target multiple nucleosomes ChIP-seq experiments should be performed to examine whether MiDAC associates with specific chromosomal regions in the context of the mitotic phenotype or specific genes in the context of the deregulated genes.

These are all good points and we agree are important experiments to further elucidate the mechanistic role of MiDAC. However, we believe these are beyond the scope of this current manuscript and would require a large amount of additional experimental work. We believe this study is a comprehensive analysis of an understudied HDAC complex that will pave the way for future work to uncover the specific mechanisms of how MiDAC functions in terms of gene regulation and localised histone acetylation.

In the current form the analysis of the KO mice adds little to the manuscript. The phenotype of MEFs is similar to the one of knock-down U2OS cells and the impact on mitosis does not affect early embryogenesis. The effects on heart development and the hematopoietic system are unclear and might be investigated in another study. In this context, loss of the MiDAC complex could be partially compensated in the early phase of development by a related HDAC complex, which might be absent in specific differentiated cell types.

We agree with the reviewer that the mechanisms through which loss of the MiDAC complex results in defects the heart and haemopoietic system are unclear. However, we believe that the knockout mice are important in the context of our paper for four reasons:

- They reveal for the first time that the MiDAC complex is essential for life and the loss cannot be compensated for by other complexes containing class I HDACs.
- The identical phenotype and gene expression changes of the two knockouts reinforces the finding that MIDEAS and DNTTIP1 are mutually interdependent.
- They also confirm that the chromosome mis-alignment phenotype of the knockdowns in cancer cell lines is due to loss of the MiDAC complex.
- Finally, the RNAseq data supports the concept that MiDAC is a gene regulatory complex. This has not been previously demonstrated.

As the reviewer suggests we agree that investigation of the causes for the effects on the heart and hematopoietic systems will best be investigated in a future study.

To address this point we have combined figures 3 and 4 so as to reduce the emphasis on this part of the manuscript. We have also spelled out in the text why the mouse

knockouts are important for our understanding of the complex and also the limitations of our current study.

In the first part the authors provide convincing data showing that the MiDAC complex is functional as deacetylase complex not only during mitosis but throughout the cell cycle. As discussed by the authors this is in contrast to previous findings and would suggest that the MiDAC complex has a more general function. Therefore it would be important to show that HDAC1 is associated with DNTTIP1 and MIDEAS during interphase.

This is a good point that was also raised by reviewer 1. We now show in a new figure (Figure 1G) that HDAC1 is associated with MIDEAS and DNTTIP1 during interphase as shown by a G1/S block, as well as in mitosis.

Reviewer #3 (Remarks to the Author):

MiDAC is one of the class I histone deacetylases. The global structure and detailed mechanisms remain unclear. As other class I histone deacetylases, MiDAC is a multi-protein complex, composed of an HDAC catalytic subunit and two scaffold subunits, MIDEAS and DNTTIP1. In this manuscript, Turnbull, et al. designed a series of functional and structural studies to characterize the MiDAC complex and explore its functions. They determined the cryo-EM structures of MiDAC at resolutions of 4-5 angstrom, revealing a unique architecture different from other class I histone deacetylases.

Overall this work was well executed with a large body of data from a combination of different tools, including cell biology, animal model and structural biology. Before formal acceptance for publication, several issues should be addressed in the revision.

1. As stated in the last sentence in Abstract, "Four copies of HDAC1 are positioned at the periphery with outward-facing active sites so as to target multiple nucleosomes suggesting...". "target multiple nucleosomes" is a hypothesis based on cryo-EM structures and not supported by any functional data in previous or this study. Therefore, the wording should also be more accurate.

We have data from our previous paper that shows that the complex binds with higher affinity to chromatin with multiple nucleosomes (Itoh et al. 2015). The architecture of the complex from the EM with outward facing active sites suggests that the complex will target multiple nucleosomes simultaneously. We have adjusted the wording.

2. In Figure 2A, the definition of y-axis, "Relative Density", should be explained in legend. The reference value for normalization in the histogram was not clearly defined. Were the column heights normalized based on the Control lane? If so, why the height of Control lane in the top panel was more than 2? In addition, in the bottom panel, the protein concentrations in MID_si1/2 were significantly less than Control lane, but the column heights are similar.

We thank the reviewer for pointing this out and apologise for any confusion. The relative density was calculated with respect to the housekeeping protein Lamin B explaining why the values for the control lane are more than 1.

With regards to the final point, this was a mistake. We had originally included the qRT-PCR results here. We later decided to move these to the supplementary material (Supp. Figures 3A-C in the revised manuscript) and replaced these with a quantification of the western blotting results. Unfortunately, the qRT-PCR graph for DNTTIP1 was not replaced with the western quantification (the graph in original figure 2A DNTTIP1 is the same as the original Supp. Figure 2B) this has now been corrected. We are grateful to the reviewer for spotting this unfortunate error.

3. The description in the second paragraph of Page 6. In Supplementary Figure 2A-B, the knock-down of MIDEAS or DNTTIP1 only suppressed the mRNA expression of their own. But in Figure 2A, the protein levels of both MIDEAS and DNTTIP1 were affected in either MIDEAS or DNTTIP1 depleting cell lines. It seems that the “mRNA expression” and “protein levels” were detected in the same manner using western blotting and the difference above accounted for the difference sampling: whole cell extract v.s. soluble nuclear fraction. This should be clarified in the revision.

We believe this confusion has been caused by the mistake with the graphs mentioned above. The protein levels in figure 2A are representative of western blotting results and both MIDEAS and DNTTIP1 protein levels were decreased in both MIDEAS and DNTTIP1 depleting cell lines. The mRNA levels depicted in supp. Figures 3 A-C (previously Supp Fig. 2) were quantified by qRT-PCR and it can be seen that in this case the siRNA's are specific in depleting their target transcript with little or no effect on the mRNA levels of the other components.

4. In Figure 3C-D, the background and contrast in these images seemed to have been manually adjusted. If the adjustment is necessary, details should be provided in the legend.

We appreciate the images look different in terms of background. This is due to the microscope used to image the sections and the H&E staining. We have re-imaged the same slides using a different microscope using the same brightness and contrast settings to remove any ambiguity that may arise.

5. What does the “c. 7 Å” in the first paragraph of Page 10 and other parts of the article mean? Please describe it upon the first usage of this term.

In UK English “c.” is an accepted abbreviation for “circa” (from Latin, meaning ‘approximately’). I believe in other European countries “ca.” is more common. We are happy to use whatever format Nature Communications prefers. For now, we have changed this to a tilde “~” as this seems to be a more internationally recognised abbreviation.

6. The structural data show a unique architecture for the MiDAC complex. However, many conclusions in the text were presented without supporting figures provided (listed below). The authors should provide a number of necessary supporting figures for better presentation of their structural data:

(1) The last sentence in Page 10, “Density for the InsP6, but not the SAHA, was clearly visible in the map.”;

Supplementary figure added (Supp. Fig. 10A)

(2) The second paragraph in Page 11, “The shorter helix in MIDEAS mediates a tight interaction with a non-polar groove between the end of the long helices of the DNTTIP1 dimer”;

Supplementary figure added (Supp. Fig. 10B)

“The SANT domains of MIDEAS and MTA1 are very similar in structure to each other and to the SANT domain of the SMRT protein that binds to HDAC32”;

Supplementary figure added (Supp. Fig. 10C)

“One of the regions of MIDEAS most similar to other ELM2-SANT domain corepressor proteins is the ELM2 specific motif which binds in an extended conformation in a conserved groove on HDAC1”.

Supplementary figure added (Supp. Fig. 10D)

(3) The first paragraph in Page 12, “The arrangement is such that the DNTTIP1 dimerisation domains are symmetrically juxtaposed, but do not directly interact.”;

We now refer to Figure 6B and Supp. Fig. 11 where this is clearly shown.

“The two S-shapes are in opposite orientations such that the four HDAC subunits are positioned at the periphery, resulting in a striking three-dimensional X-shaped complex (Supplementary Figure 7H)”. “opposite orientations” should be clearly displayed.

We now refer to Figure 6B and Supp. Fig. 11 where this is clearly shown and have expanded the description to clarify our meaning.

(4) The second paragraph in Page 12, “The tetramerisation interface between the two dimers is mediated by interactions between the SANT domains”;

We now refer to Figure 6C and have highlighted the SANT domains

“Residues Lys849, Tyr874 and Gln877 contribute to the interface between the two neighbouring SANT domains”;

Supplementary figure added (Supp. Fig. 10D)

“These residues are not conserved in MTA1”;

We now refer to figure 5E and have labelled these residues

“Interestingly, Lys839 and Lys843 appear to be in a position to interact with the inositol phosphate that is bound to the neighbouring dimer, although there is no apparent density for the side chains”;

Supplementary figure added (Supp. Fig. 10 D & E)

“This may explain why it was only possible to obtain a high-resolution map of the tetramer in the presence of inositol phosphate (7 Å vs 4.5 Å)”. The image processing for “the tetramer in without inositol phosphate” should be described in Method session. Related figure should also be provided.

We now show the data processing for all datasets in Supp. Figs. 7 & 8

“A further obvious effect of the InsP6 was to significantly improve the local resolution of the core of the HDAC1, consistent with the proposed mechanism of activation”.

Supplementary figure added (Supp. Fig. 10F)

(5) The third paragraph in Page 12, “In addition to the interactions between the SANT domains it appears that the residues Nterminal to the DNTTIP1 dimerisation domain are crossing over to interact with the neighbouring tetramer”.

Supplementary figure added (Supp. Fig. 10G)

(6) The last paragraph in Page 12, “We also analysed cryo-EM grids of complexes containing full-length DNTTIP1 including the DNA-binding domain (Supplementary Figure 7A-D)”. The cited figures were incorrect.

This has been corrected

(7) The first paragraph in Page 16, “Interestingly, the tetrameric arrangement brings the inositol phosphate binding sites on the two dimers in close proximity such that two conserved basic residues in the SANT domain of one MIDEAS are able to interact across the tetramerisation interface with the inositol phosphate in the adjacent dimer”.

We now refer to figure 6C and Supp. Fig. 10 D & E

7. In Supplementary Figure 5, cryo-EM maps of the intermediate processing steps, especially for the results of 3D classification, should be displayed here.

We now show the data processing for all datasets in Supp. Figs. 7 & 8

8. Judged by Supplementary Figure 6D, there seems to be a preferred orientation problem for the tetramer complex. Please provide a figure panel displaying the angular distribution of the final refinement.

The reviewer is correct, both the dimer and tetramer suffer from preferential orientation. This is now shown clearly in Supp. Figs. 9C & 9F.

9. The Data collection and Image processing for the data from grafix cross-linking, and

gentle cross-linking samples without DNTTIP1 truncation could be included in Method section.

We have added more detail of the data processing in the supplementary figures 7 & 8. This covers four datasets deriving from grids and samples prepared in different ways. We have not included the Grafix data set as this did not inform the direction of our experiments.

10. As stated in the last paragraph of Page 11, “The most ordered regions are the DNTTIP1 dimerisation domain and the core of HDAC1 (Figure 7A & B and Supplementary Figure 6)”. That means the regions of MIDEAS are less ordered? Since MIDEAS is located between DNTTIP1 and HDAC1, not the most peripheral, does the lower resolution correlate with its functional dynamics for the complex?

We think that the differences in resolution are insufficient to safely draw in conclusions about the functional dynamics of the complex. We have weakened the statement in the text.

Have the authors tried to perform local 3D classification focused on the MIDEAS region to separate different conformations?

MIDEAS is probably too small for local classification. We have tried local 3D classification focussed on one monomer of the tetramer. We have tried using the Multibody refinement feature in Relion. We have also tried symmetry expansion to refine the structure as the equivalent of monomers. None of these approaches significantly improved the resolution.

REVIEWERS' COMMENTS:

Reviewer #1 (Remarks to the Author):

Most of the concerns raised previously have been taken into account and the manuscript has now been much improved.

I can therefore recommend this manuscript for publication.

Saadi Khochbin

Reviewer #3 (Remarks to the Author):

Most of the issues have been well responded. Overall, this is an excellent paper.

1. As a response to question 6-(2) of the third reviewer, Supp. Fig. 10B was provided. Some labels could be added in this panel to improve the clarity, including labels for “the shorter helix”, “a non-polar groove” and “the end of the long helices”.

3. As to the question 6-(4), the “7Å vs 4.5 Å” looks like “7.7 Å vs 4.5 Å” or “8 Å vs 4.5 Å” according to the added Supp. Fig. 8.

REVIEWERS' COMMENTS:

Reviewer #1 (Remarks to the Author):

Most of the concerns raised previously have been taken into account and the manuscript has now been much improved. I can therefore recommend this manuscript for publication.

Saadi Khochbin

Reviewer #3 (Remarks to the Author):

Most of the issues have been well responded. Overall, this is an excellent paper. 1. As a response to question 6-(2) of the third reviewer, Supp. Fig. 10B was provided. Some labels could be added in this panel to improve the clarity, including labels for “the shorter helix”, “a non-polar groove” and “the end of the long helices”. 3. As to the question 6-(4), the “7Å vs 4.5 Å” looks like “7.7 Å vs 4.5 Å” or “8 Å vs 4.5 Å” according to the added Supp. Fig. 8.

We have made these changes